# Revisiting End-to-End Learning with Slide-level Supervision in Computational Pathology

**Wenhao Tang**[*1,2]    **Rong Qin**[*2]    **Heng Fang**[3]    **Fengtao Zhou**[4]    **Hao Chen**[4]

**Xiang Li**[†1,2]                    **Ming-Ming Cheng**[†1,2]

[1]Nankai International Advanced Research Institute (Shenzhen Futian)
[2]VCIP, School of Computer Science, Nankai University
[3]Huazhong University of Science and Technology
[4]The Hong Kong University of Science and Technology

[*]Equal Contribution. [†]Corresponding author.

## Abstract

Pre-trained encoders for offline feature extraction followed by multiple instance learning (MIL) aggregators have become the dominant paradigm in computational pathology (CPath), benefiting cancer diagnosis and prognosis. However, performance limitations arise from the absence of encoder fine-tuning for downstream tasks and disjoint optimization with MIL. While slide-level supervised end-to-end (E2E) learning is an intuitive solution to this issue, it faces challenges such as high computational demands and suboptimal results. These limitations motivate us to revisit E2E learning. We argue that prior work neglects inherent E2E optimization challenges, leading to performance disparities compared to traditional two-stage methods. In this paper, we pioneer the elucidation of optimization challenge caused by sparse-attention MIL and propose a novel MIL called ABMILX. ABMILX mitigates this problem through global correlation-based attention refinement and multi-head mechanisms. With the efficient multi-scale random patch sampling strategy, an E2E trained ResNet with ABMILX surpasses SOTA foundation models under the two-stage paradigm across multiple challenging benchmarks, while remaining computationally efficient ($< 10$ RTX3090 GPU hours). We demonstrate the potential of E2E learning in CPath and calls for greater research focus in this area. The code is here.

## 1   Introduction

Computational pathology [15, 53, 14] (CPath) is an interdisciplinary field that combines pathology, gigapixel image analysis, and computer science to develop computational methods for analyzing and interpreting pathological images (whole slide images, WSIs or slides). This field leverages advanced algorithms, machine learning, and artificial intelligence techniques to assist pathologists in tasks such as cancer sub-typing [25, 66], grading [6], and prognosis [63, 65]. Due to clinical demands and the challenge of pixel-level annotation in gigapixel pathological images, CPath typically focuses on slide-level learning. However, analyzing such gigapixel images in slide-level presents significant challenges in terms of efficiency and performance.

To address these challenges, Campanella et al. [7] proposed a two-stage paradigm based on multiple instance learning (MIL) [41], allowing efficient WSI analysis without fine-grained annotations. This

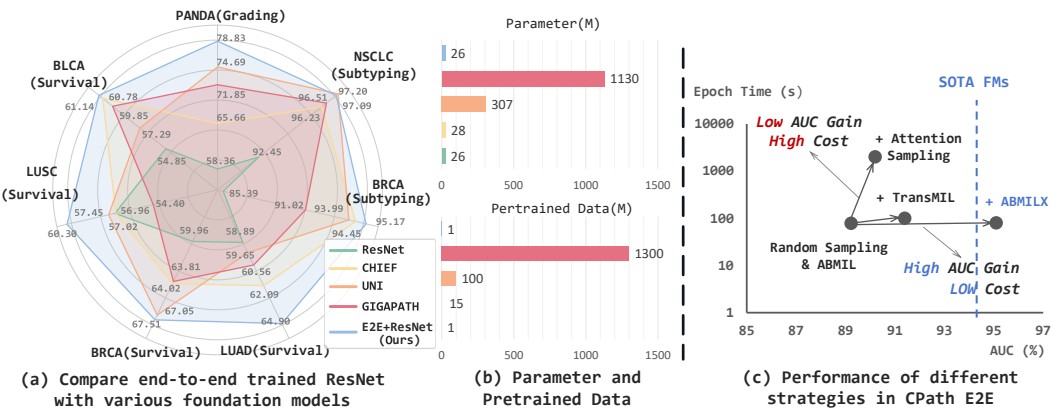

Figure 1: (a,b) We compare E2E trained ResNet with various foundation models using two-stage paradigm in terms of performance, model size, and pretraining data. This demonstrates the performance potential of E2E learning for computational pathology under low computational budget. (c) Compared to sampling strategies, different MILs have a more significant impact and lower cost on E2E learning.

approach first divides each WSI (a bag) into thousands of image patches (instances). Pretrained encoders extract offline instance features, which are then aggregated into bag features by a sparse-attention MIL model, ultimately leading to slide prediction. By operating in the latent space rather than images, this paradigm enables slide-level supervised training within reasonable memory constraints. However, its performance heavily depends on the quality of offline features [10]. To improve offline feature quality, a series of pathology foundation models [11, 62, 24, 64] (FMs) like UNI [11] and GigaPath [64] have been developed. As shown in Figure 1, despite scaling data volume to 170K WSIs (>200TB) and model size over 1B, these approaches still perform unsatisfactorily on specific tasks. We attribute this to the lack of unified optimization in the two-stage paradigm, resulting in encoders with insufficient adaptation of downstream task and disjoint optimization with MIL models.

End-to-end supervised learning with joint encoder and MIL at the slide level (E2E learning) offers a fundamental solution, enabling efficient downstream data utilization and task-specific encoder learning. However, due to prohibitive computational costs and suboptimal performance, this area remains underexplored. Existing works [50, 9, 61] typically employ patch sampling to maintain a reasonable computational budget, focusing on improving sampling quality to enhance performance. However, previous work *overlooked the optimization challenges introduced by MIL in E2E learning, resulting in limited performance improvements*. The results in Figure 1(c) show that complex sampling strategies incur significant time costs with minimal performance gains. And different MILs significantly impact E2E training. Specifically, E2E learning with sparse-attention MIL performs poorly, falling below SOTA MIL methods using offline features extracted by ResNet-50 (R50) and significantly underperforming SOTA FMs. As shown in Figure 2, sparse attention is crucial for CPath, enabling models to focus on key regions from thousands of patches and performs increasingly well with superior features. However, we suggest that it can also disrupt the encoder in E2E learning due to its insufficient consideration of discriminative regions and potential extreme focus on redundant ones. Poor features further affect the accuracy of attention in the next iteration, leading to deteriorating iterations and compromising the entire optimization process.

To retain the benefits of sparse attention while mitigating its induced optimization challenges in E2E learning, we propose ABMILX, a novel MIL model based on the widely used ABMIL [25]. ABMILX incorporates multi-head attention mechanism to capture diverse local attention from different feature subspaces, and introduces a global attention plus module that leverages patch correlations to refine local attention. Both modules help the encoder learn more discriminative regions and avoid excessive focus on redundant areas. Furthermore, we adopt simple but effective multi-scale random patch sampling to incorporate multi-scale information while reducing E2E learning computational costs. Our E2E learning framework achieves significant performance improvements (e.g., +20% accuracy on PANDA) while maintaining computationally efficient (< 10 RTX3090 GPU hours on TCGA-BRCA). The main contributions can be summarized as follows:

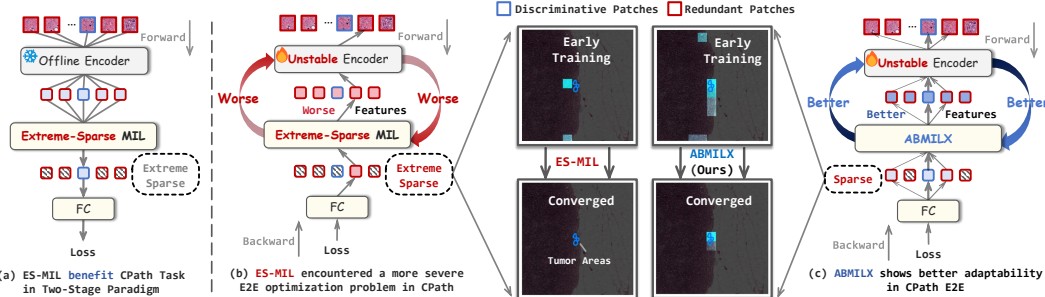

Figure 2: In E2E learning, MIL can be viewed as an soft instance selector that iteratively optimizes with the encoder. The encoder outputs instance features to MIL for attention-based aggregation and receives the instance gradients for optimization. The attention from MIL affects the gradients of different instance features, leading to selective learning of patches by the encoder. In contrast to two-stage learning approaches, the commonly used excessively sparse attention makes the encoder optimization overfitted on limited discriminative regions and vulnerable to redundant ones. Worse features further affect the accuracy of selection, compromising the optimization loop.

- We revisit slide-level supervised E2E learning for CPath and pioneer the identification of optimization challenges. We show that E2E learning with slide-level supervision and its optimization collapse risks from the sparse attention of MIL deserve more attention.
- To address E2E learning optimization challenges while maintaining sparse attention, we propose the ABMILX model. By incorporating multi-head attention mechanisms and global correlation based attention plus modules, it significantly improves performance.
- We propose a slide-level supervised E2E learning pipeline based on multi-scale random patch sampling. It keeps a reasonable computational budget and introduces multi-scale information. Within this pipeline, an E2E trained ResNet with ABMILX surpasses the SOTA FMs under two-stage frameworks across multiple challenging benchmarks. This pioneerly demonstrates the potential of E2E learning in CPath.

## 2 Related Works

**Computational Pathology.** The advent of WSI in computational pathology (CPath) has revolution-ized approaches to cancer diagnosis and prognosis by furnishing a comprehensive, high-resolution view of tissue specimens [15, 53, 14]. Due to processing gigapixel images is computationally in-tensive, traditional CPath methods have adopted a two-stage paradigm to prioritize efficiency [38]. In the first stage, an offline feature extractor, typically pre-trained in general datasets [38, 21] or pathology datasets [11, 62, 24, 64], is employed to encode tissue patches into features. In the subsequent stage, MIL are applied to aggregate these features for slide-level prediction. Most re-search [26, 31, 51, 55, 58, 33, 54, 68, 32, 57] has focused on refining MIL stage with advanced aggregation mechanisms, notably the use of sparse attention. MIL model computes attention scores for each patch and aggregates only the most informative ones [25], thereby reducing noise and en-hancing slide-level prediction accuracy in WSIs with scattered key histological features. Some studies have sought to better exploit information contained in WSIs by directly extracting supplementary visual cues from the entire slide [52, 19]. Others have refined the extracted features to better match the dataset through either multi-stage feature extractor fine-tuning [30] or online instance feature re-embedding [56] to more precisely tailor the extracted features to the dataset. However, current two-stage approaches rely on pretrained offline feature extractors that are not jointly optimized with the MIL model, potentially resulting in features that inadequately capture the complex nuances of pathology data in WSIs [30]. In this context, E2E approaches have emerged as a promising paradigm.

**E2E Learning in Computational Pathology.** E2E learning, which jointly optimizes feature extrac-tion and prediction from WSIs in CPath, offers a more adaptive encoder that enhances the discrim-inability of representations. However, high computational costs and unsatisfactory performance have hindered systematic research in this area, with existing E2E CPath methods falling into instance-level and slide-level supervised approaches. Instance-level supervised methods [13, 39, 46, 47] simplify processing by training encoders with pseudo-labels for individual patches rather than slide labels.

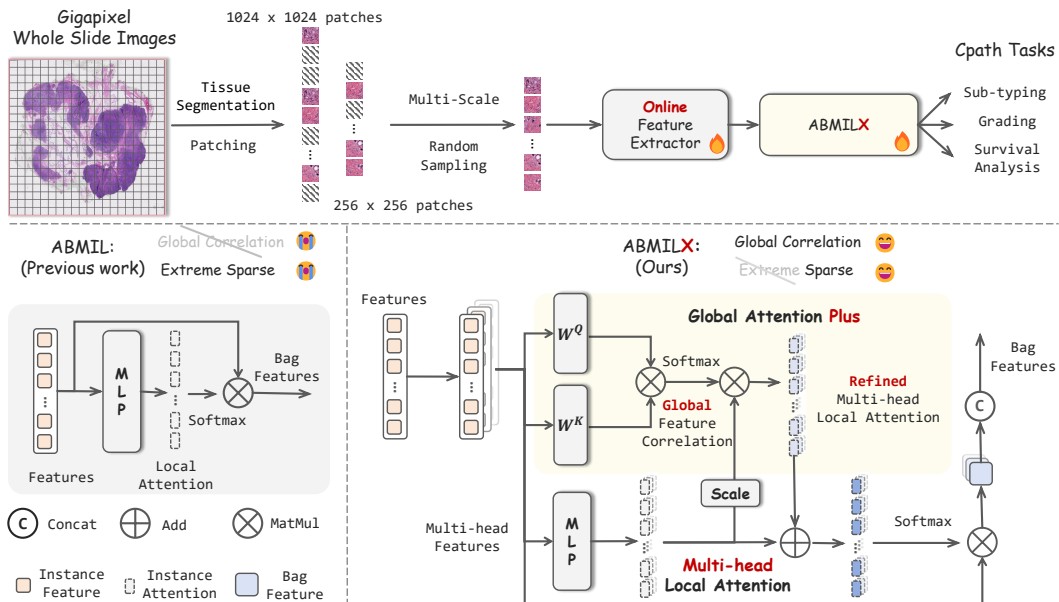

Figure 3: Overview of the proposed E2E training pipeline and ABMILX. ABMILX introduces multi-head local attention to address the extreme sparsity issue in ABMIL [25], which hinders E2E optimization. Furthermore, ABMILX refines the local attention using global feature correlations via the attention plus. This encourages the model to focus on task-specific regions during E2E learning.

They neglect the crucial inter-patch context required for clinical analysis [7, 40, 23], making them a compromise rather than a fully end-to-end solution. In contrast, slide-level supervised methods analyze entire slide to preserve contextual interdependencies and deliver a comprehensive, clinically relevant analysis. It is broadly classified into two main categories. The first group leverages memory-efficient architectures and model-parallel techniques to enable E2E learning on gigapixel images [43, 8, 17, 59, 34, 60], yet it still demands substantial computational resources (e.g., 64 V100 GPUs [60]) and fails to deliver satisfactory outcomes [42]. Alternatively, another group alleviates the efficiency challenges with data sampling to train on selected subsets [50, 9, 61]. This approach focuses on diverse sampling mechanisms such as clustering-based [50] and attention-based [9, 61, 12] methods, to identify key regions from slide. These complex sampling also give rise to iterative [50, 9] and multi-stage [30] pipelines. In summary, although current E2E methods are complex and computationally intensive, they still underperform FM-based two-stage methods. This performance gap can be attributed to overlooked optimization challenges caused by MIL. In this paper, we pioneer the proposition that the optimization challenges are the performance bottleneck of E2E methods.

## 3 Methodology

### 3.1 Slide-level Supervised End-to-End Learning

The lack of encoder adaptation in the two-stage paradigm limits the feature specificity on CPath tasks, thereby calling for slide-level supervised E2E training to jointly optimize the MIL model and the encoder. The upper of Figure 3 shows the overall E2E learning pipeline of our method, which consists of multi-scale random instance sampling, instance feature encoder, ABMILX, and task head. Specifically, given the target number of sampled instance $s$ and a slide $\boldsymbol{X}$, a instance subset $\boldsymbol{L}$ could be collected through our multi-scale random instance sampling strategy $\mathcal{V}(\cdot)$ as encoder input to avoid massive training cost, $\boldsymbol{L} = \mathcal{V}(s, \boldsymbol{X})$. The $i$-th instance $\boldsymbol{l}_i$ is embedded into an instance feature $\boldsymbol{e}_i \in \mathbb{R}^D$ by an encoder, $\boldsymbol{e}_i = \mathcal{F}_\theta(\boldsymbol{l}_i)$. $\mathcal{F}(\cdot)$ denotes the mapping functions of any encoder and the $\theta$ denotes the corresponding learnable parameters. After that, the features of all sampled instances $\boldsymbol{E} = \{\boldsymbol{e}_1, \cdots, \boldsymbol{e}_i, \cdots, \boldsymbol{e}_s\}$ will be aggregated through our proposed ABMILX, $\boldsymbol{Z} = \Gamma_\phi(\boldsymbol{E})$. The $\Gamma(\cdot)$ denotes the mapping functions of ABMILX and the $\phi$ denotes the corresponding learnable parameters. Then slide features $\boldsymbol{Z}$ are inputted into a task head $\mathcal{H}_\eta$ for the slide-level prediction $\hat{y}$,

$\hat{y} = \mathcal{H}_\eta(Z)$. Finally, we only utilize the slide-level ground truth $y$ and the $\hat{y}$ to joint optimize the aforementioned modules through task loss function $\mathcal{L}$:

$$\{\hat{\theta}, \hat{\phi}, \hat{\eta}\} \leftarrow \arg\min_{\theta, \phi, \eta} \sum_{i=1}^{n} \mathcal{L}(y_i, \hat{y}_i), \tag{1}$$

where $n$ denotes the number of slides in train set, while $\hat{\theta}$, $\hat{\phi}$, and $\hat{\eta}$ are the final parameters of encoder, MIL, and task head, respectively. Considering that E2E learning allows the attention from MIL to affect the instance gradients backpropagatd to encoder, the key insight of our method is to guide the encoder to learn task-specific discriminative regions through our proposed ABMILX.

**Multi-scale Random Instance Sampling.** The sampling stage aims to take a subset from massive instances for training, thereby reducing the cost of E2E learning. The sampling methods generally fall into random and selective sampling [50, 9, 61, 27]. The latter focuses on traversing the slide to obtain high-value instance samples, which significantly increases training time and heavily relies on the evaluation model [49, 30, 12]. In this paper, we introduce a multi-scale random instance sampling (MRIS) method to maintain low training cost while leveraging multi-scale instances to capture information at different granularities. Specifically, given a multi-scale set $\{I_1, \cdots, I_j, \cdots, I_t\}$, we adopt a sampling ratio set $\{\sigma_1, \cdots, \sigma_j, \cdots, \sigma_t\}$ to obtain the number $\hat{s}_j$ of $\hat{I}_j$ for sampling:

$$\hat{\boldsymbol{L}}_j = \mathcal{V}_\mathcal{S}(I_j, \hat{s}_j, \boldsymbol{X}), \hat{s}_j = \lceil s \times \sigma_j \rceil, \tag{2}$$

where $\mathcal{V}_\mathcal{S}(\cdot)$ denotes the function of vanilla random sampling. It is notable that we set $\sum_{j=1}^{t}(\sigma_j) = 1$ to ensure that $\sum_{j=1}^{t}(\hat{s}_j) = s$. We resize the sampled instances of different scales and context extents $\{\hat{\boldsymbol{L}}_1, \cdots, \hat{\boldsymbol{L}}_j, \cdots, \hat{\boldsymbol{L}}_t\}$ to a unified resolution and merge them as the final sampling set $\boldsymbol{L}$. On the one hand, multi-scale sampling simulates the multi-scale perspective of pathologists during diagnosis and improves the CPath performance of our method. On the other hand, the unified resolution for different context avoids the additional cost and maintains parallel training, while remaining the different scale perspectives of original instance. Appendix C.4 give more details about sampling.

## 3.2 ABMILX for Effective End-to-End Learning

Sparse-attention MIL that relies on local instance features, such as the most representative AB-MIL [25], could avoid key regions being overwhelmed by redundant instances and performs increasingly well with superior features. However, we demonstrate that the sparse attention will introduce interference risks in E2E learning and bring suboptimal performance. The risks primarily stem from the insufficient consideration of discriminative instances and excessive focus on redundant ones. In this paper, we propose ABMILX, which consists of a multi-head local attention and a global attention plus module to mitigate the optimization risks from both local and global perspectives. It also maintains the sparse characteristic to effectively collaborate with the fine-tuned encoder.

**Multi-head Local Attention.** Considering that the false attention from under-converged MIL usually exhibit a random distribution, we propose a multi-head local attention module (MHLA) to directly suppress the excessive focus on redundant instances while improving the attention for the discriminative ones. Specifically, we divide the features of all sampled instances $\boldsymbol{E}$ into $m$ head features $\{\boldsymbol{H}^1, \cdots, \boldsymbol{H}^j, \cdots, \boldsymbol{H}^m\}$, where $\boldsymbol{H}^j \in \mathbb{R}^{s \times \lceil D/m \rceil}$. Within each head, the head features are input into a shared MLP to compute the corresponding attention, $\boldsymbol{A}^j = \text{MLP}(\boldsymbol{H}^j)$. The $\boldsymbol{A}^j \in \mathbb{R}^{s \times 1}$ denotes the local attention vector of the $j$-th head, which possesses sparse characteristic important in CPath tasks. In the E2E learning, the separate voting from multiple heads allows to reduce the excessive focus on redundant instances, while the attention from different feature subspaces helps to provide a more comprehensive view on discriminative instances. Finally, we aggregate the features within each head through $\boldsymbol{A}^j$ to obtain the head-level slide features $(\boldsymbol{Z}^1, \cdots, \boldsymbol{Z}^j, \cdots, \boldsymbol{Z}^m)$, which are then concatenated as the final slide feature :

$$\boldsymbol{Z} \in \mathbb{R}^{1 \times D} = \text{Concat}(\boldsymbol{Z}^1, \cdots, \boldsymbol{Z}^j, \cdots, \boldsymbol{Z}^m), \boldsymbol{Z}^j = \text{Softmax}(\mathcal{G}(\boldsymbol{A}^j))^T \boldsymbol{H}^j, \tag{3}$$

where $\mathcal{G}(\cdot)$ denotes the mapping function of our global attention plus module. It aims at further refining $\boldsymbol{A}^j$ through propagating sparse attentions from discriminative instances to their similar instances for better feature aggregation and optimization. Compared to directly averaging the head attention and aggregating the whole instance features, head-level aggregation enables MIL to obtain more diverse representations from different feature subspaces.

**Global Attention Plus Module.** Tissues with similar pathological characteristics typically exhibit highly similar morphology, leading to a higher correlation among corresponding instance features. Therefore, besides directly enhancing attention from the local instance perspective, we propose an global attention plus module (A+) to leverage the global correlations for attention refinement, which could indirectly improve the focus for the discriminative instances while suppressing the redundant ones. It propagates $A^j$ between similar instances to obtain a the global sparse attention and then combines it with $A^j$, thereby correcting the local sparse attention from MHLA. When integrating the MHLA with the A+ module, we first share A+ module across different heads to obtain the refined head-attention by computing a similarity matrix $U^j$, respectively, and then perform feature aggregation within each head for the refined head-level slide features as mentioned in Eq 3:

$$\mathcal{G}(A^j) = A^j + \alpha \cdot U^j A^j = A^j + \alpha \cdot \text{Softmax}(\frac{Q^j K^{jT}}{\sqrt{\lceil D'/m \rceil}}) A^j, \qquad (4)$$

where $Q_i^j = H^j W^q, K^j = H^j W^k$. The $W^q \in \mathbb{R}^{\lceil D/m \rceil \times \lceil D'/m \rceil}$ and $W^k \in \mathbb{R}^{\lceil D/m \rceil \times \lceil D'/m \rceil}$ are both the linear transforms. To preserve the sparsity, we introduce a shortcut branch with a learnable scaling factor $\alpha$ that adaptively combines global sparse attention $U^j A^j$ and the original local one.

The propagation weight of the $i$-th instance, denoted as $P(i)$, is defined as the sum of its influence on all instances. In classic transformer-based methods, the propagation weights $P_{trans}$ is determined by only the similarity matrix $U^j$. *However, for the global sparse attention introduced in ABMILX, the weights $P_{abx}$ is also significantly affected by the original sparse head attention value $A^j$:*

$$P_{trans}(i) = \sum_{k=1}^{s} U_{k,i}^j \xrightarrow{A^j} P_{abx}(i) = A_k^j \sum_{k=1}^{s} U_{k,i}^j. \qquad (5)$$

Therefore, ABMILX utilizes the $A^j$ as prior distribution to grant sparse discriminative instances with higher propagation weights to find more potential instances while suppressing the normal ones. *More theoretical analysis about ABMILX is available in Appendix A.*

### 3.3  Sparse Attention Analysis in E2E Learning

To intuitively analyze the effect of sparse attention on E2E training, we quantitative the sparsity of different MILs by the proportion of activated patches. Sparsity is statistically derived from the CAMELYON dataset [3]. Moreover, the right figure visualizes attention scores (bottom) and corresponding distribution (middle) of MILs during training. We demonstrate the following: (1) In E2E optimization, extreme sparsity causes ABMIL to overlook discriminative regions while over-focusing on redundant ones, leading to worst performance. (2) Although the global attention of TransMIL eliminates this extreme sparsity and covers some discriminative regions, it is also largely distracted by the redundant ones, which also brings limited accuracy gains. (3) In contrast, both MHLA and A+ make ABMILX maintains reasonably sparse attention, which considers most of the discriminative regions while maintaining low attention to normal patches. Benefited from them, ABMILX achieves the best performance in different C-Path tasks. Besides, learnable $\alpha$ also helps adaptively adjusting the sparsity and brings more accuracy gains. *More experiments and analysis about the affect of different MILs in E2E learning are available in Sec. 4.3 and Appendix C.2.*

| Different MILs in E2E | Sparsity | Sub.↑ | Surv.↑ |
|---|---|---|---|
| ABMIL | 80 | 89.23 | 62.70 |
| TransMIL | 13 | 91.44 | 63.42 |
| MHLA ($\alpha = 0$) | 61 | 91.58 | 63.80 |
| MHLA&A+ ($\alpha = 1$) | 29 | 92.84 | 65.49 |
| MHLA&A+ (learnable $\alpha$) | 36 | 93.97 | 67.78 |

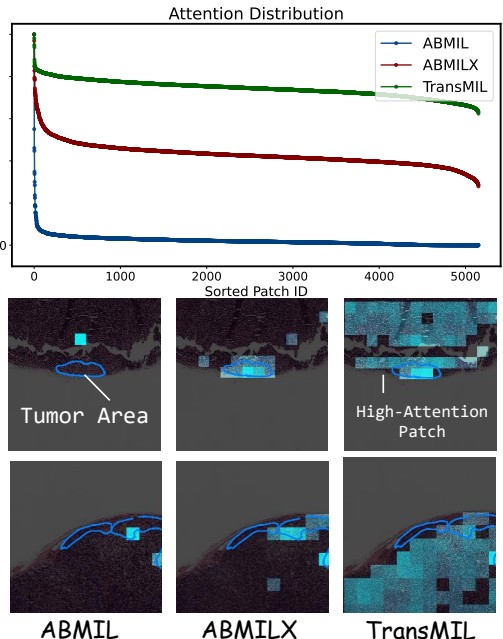

Table 1: Sub-typing results on two main datasets and training cost of different CPath methods.

| Encoder | Method | E2E | Pretraining Data | #Parameter | FLOPs | TCGA-BRCA | TCGA-NSCLC |
|---|---|---|---|---|---|---|---|
| ResNet-50 | ABMIL [25] | ✗ | ImageNet-1K [16] ~ 1M Data | 26M+0.66M | ~2.12T | $83.80_{\pm6.55}$ | $92.32_{\pm2.68}$ |
| | CLAM [38] | | | 26M+0.79M | | $85.86_{\pm6.43}$ | $92.28_{\pm2.69}$ |
| | TransMIL [48] | | | 26M+2.67M | | $88.52_{\pm5.44}$ | $92.49_{\pm2.66}$ |
| | DSMIL [29] | | | 26M+0.87M | | $85.68_{\pm6.06}$ | $91.12_{\pm3.04}$ |
| | WIKG [33] | | | 26M+1.71M | | $88.37_{\pm5.35}$ | $92.57_{\pm2.53}$ |
| | RRTMIL [56] | | | 26M+2.70M | | $89.35_{\pm5.41}$ | $94.43_{\pm2.16}$ |
| | 2DMamba [67] | | | 26M+2.27M | | $87.22_{\pm5.30}$ | $95.21_{\pm2.07}$ |
| CHIEF [62] | ABMIL [25] | ✗ | Slide-60K ~ 15M Data | 27M+0.66M | ~2.24T | $91.09_{\pm4.71}$ | $96.22_{\pm1.67}$ |
| | TransMIL [48] | | | 27M+2.67M | | $91.41_{\pm3.95}$ | $96.39_{\pm1.73}$ |
| | CHIEF [62] | | | 27M+1.05M | | $91.43_{\pm4.52}$ | $96.84_{\pm1.45}$ |
| | RRTMIL [56] | | | 307M+2.70M | | $92.49_{\pm4.21}$ | $97.00_{\pm1.41}$ |
| | 2DMamba [67] | | | 27M+2.27M | | $91.88_{\pm4.11}$ | $96.93_{\pm1.60}$ |
| UNI [11] | ABMIL [25] | ✗ | Mass-100K [11] ~ 100M Data | 307M+0.66M | ~31T | $94.05_{\pm3.49}$ | $97.04_{\pm1.60}$ |
| | TransMIL [48] | | | 307M+2.67M | | $93.33_{\pm3.50}$ | $97.27_{\pm1.58}$ |
| | RRTMIL [56] | | | 307M+2.70M | | $94.61_{\pm3.18}$ | $\mathbf{97.88}_{\pm1.18}$ |
| | 2DMamba [67] | | | 307M+2.27M | | $93.08_{\pm4.20}$ | $97.14_{\pm1.48}$ |
| GIGAP [64] | ABMIL [25] | ✗ | **Slide-170K ~1.3B Data** | 1134M+0.66M | ~114T | $94.39_{\pm3.43}$ | $96.54_{\pm1.66}$ |
| | TransMIL [48] | | | 1134M+2.67M | | $93.97_{\pm3.88}$ | $97.61_{\pm1.23}$ |
| | GIGAP [64] | | | 1134M+86M | | $93.72_{\pm3.43}$ | $97.53_{\pm1.19}$ |
| | RRTMIL [56] | | | 1134M+2.70M | | $94.82_{\pm3.63}$ | $97.63_{\pm1.20}$ |
| | 2DMamba [67] | | | 1134M+2.27M | | $93.84_{\pm3.94}$ | $96.87_{\pm1.65}$ |
| ResNet-50 | Best-of-two-stage | ✗ | ImageNet-1K [16] ~ 1M Data | 26M+2.70M | 2.12T | $89.35_{\pm5.41}$ | $95.21_{\pm2.07}$ |
| ResNet-18 | C2C [49] | ✓ | | 12M+0.79M | 0.93T | 91.13 +1.78 | 95.92 +0.71 |
| ResNet-50 | FT [30] | ✓ | | 26M+0.79M | 2.12T | 86.48 −2.87 | 94.67 −0.54 |
| ResNet-18 | ABMILX (ours) | ✓ | | **12M+0.80M** | **0.93T** | **93.97** +4.62 | **97.09** +1.88 |
| ResNet-50 | ABMILX (ours) | ✓ | | 26M+0.80M | 2.12T | **95.17** +5.82 | 97.06 +1.85 |

## 4 Experiment

### 4.1 Datasets and Evaluation Metrics

We use **PANDA [6]**, **TCGA-BRCA**, and **TCGA-NSCLC** to evaluate the performance in cancer grading and sub-typing tasks. For cancer prognosis, we use **TCGA-LUAD**, **TCGA-BRCA**, **TCGA-BLCA** to evaluate performance on the survival analysis task. For external validation, we use **CPTAC-LUAD**, **CPTAC-LUSC** to evaluate the generalization ability. For cancer grading, we evaluate model performance using top-1 accuracy (Acc.). And area under the ROC curve (AUC) is used for sub-typing. For survival analysis, we employ the concordance index (C-index) [20]. To ensure robust statistical evaluation, we conducted a 1000-time bootstrapping evaluation and report the mean and 95% confidence interval. Please refer to Appendix B for more details.

### 4.2 Main Results

**Comparison Methods.** We compare several classical and latest MIL aggregators based on ResNet encoders [25, 38, 48, 29, 33, 56, 67]. Furthermore, we evaluate against three SOTA pathology FMs: UNI [11], CHIEF [62], and GigaPath (GIGAP) [64]. Following their settings, we employ ABMIL and TransMIL as aggregators. We also compare pre-trained aggregators of CHIEF and GIGAP. C2C [49] and FT [30] are E2E methods that adopt clustering-based and attention-based samplings, respectively.

**Foundation Model Dominate Two-Stage but Cost More.** Two-stage algorithms are limited by offline feature. Specifically, in grading task (Table 2), the best-performing MIL with the R50 shows a 12% accuracy gap compared to UNI with ABMIL. This performance difference is also observed in other tasks (Table 1), with gaps of 5% and 2% on BRCA-subtyping and BRCA-survival, respectively. With FM features, the superior performance of ABMIL compared with advanced methods further highlights the importance of sparse attention in CPath. However, these significant improvements come at a considerable cost. Pretraining pathology FMs demands vast amounts of data, which are difficult to acquire and share publicly. For example, UNI uses 100 million patches from approximately 100,000 slides for pretraining, while publicly available datasets typically contain fewer than 1,000 slides. The resources required by large models (e.g., GIGAP uses 3,072 A100 GPU hours) are also huge. Furthermore, the performance of FMs does not scale proportionally with increasing data and model size. Specifically, the most expensive GIGAP lags behind UNI by 3% on the PANDA. Large FMs have not achieved the same impressive performance on PANDA and BRCA as they did on

Table 2: Performance comparison across ISUP grading (PANDA) and survival analysis. OOM denotes Out-of-Memory in 24GB-3090.

| Encoder | Method | E2E | PANDA (Acc.↑) | LUAD | BRCA | BLCA |
|---|---|---|---|---|---|---|
| ResNet-50 | ABMIL [25] | ✗ | $58.89_{\pm0.80}$ | $59.56_{\pm8.6}$ | $64.93_{\pm9.1}$ | $55.01_{\pm7.9}$ |
| | CLAM [38] | | $59.45_{\pm2.18}$ | $59.79_{\pm8.7}$ | $62.90_{\pm9.4}$ | $55.78_{\pm8.0}$ |
| | TransMIL [48] | | $56.42_{\pm2.14}$ | $64.15_{\pm8.1}$ | $59.15_{\pm10.1}$ | $56.96_{\pm8.4}$ |
| | DSMIL [29] | | $61.24_{\pm2.26}$ | $61.70_{\pm8.6}$ | $61.96_{\pm9.5}$ | $56.22_{\pm8.2}$ |
| | WIKG [33] | | $62.72_{\pm2.15}$ | OOM | OOM | OOM |
| | RRTMIL [56] | | $61.97_{\pm2.17}$ | $62.19_{\pm8.4}$ | $63.03_{\pm10.2}$ | $60.78_{\pm8.2}$ |
| | 2DMamba [67] | | $61.56_{\pm2.18}$ | $61.41_{\pm7.0}$ | $61.94_{\pm8.9}$ | $54.98_{\pm6.8}$ |
| CHIEF [62] | ABMIL [25] | ✗ | $65.66_{\pm2.13}$ | $62.09_{\pm8.8}$ | $64.02_{\pm9.0}$ | $60.78_{\pm8.6}$ |
| | TransMIL [48] | | $60.89_{\pm2.23}$ | $\mathbf{65.55}_{\pm8.3}$ | $61.46_{\pm9.4}$ | $58.83_{\pm8.3}$ |
| | CHIEF [62] | | $64.24_{\pm2.12}$ | $60.29_{\pm8.1}$ | $\mathbf{67.95}_{\pm8.5}$ | $59.63_{\pm8.3}$ |
| | RRTMIL [56] | | $69.73_{\pm0.67}$ | $63.82_{\pm8.6}$ | $67.30_{\pm9.1}$ | $\mathbf{61.39}_{\pm7.9}$ |
| | 2DMamba [67] | | $72.49_{\pm1.96}$ | $60.57_{\pm8.4}$ | $64.30_{\pm9.4}$ | $59.84_{\pm8.4}$ |
| UNI [11] | ABMIL [25] | ✗ | $74.69_{\pm2.11}$ | $59.65_{\pm8.8}$ | $67.05_{\pm10.2}$ | $57.29_{\pm8.6}$ |
| | TransMIL [48] | | $68.06_{\pm2.05}$ | $60.43_{\pm9.4}$ | $62.76_{\pm10.5}$ | $60.45_{\pm8.6}$ |
| | RRTMIL [56] | | $74.93_{\pm0.53}$ | $61.64_{\pm8.8}$ | $66.91_{\pm10.1}$ | $61.07_{\pm8.3}$ |
| | 2DMamba [67] | | $76.37_{\pm2.07}$ | $61.05_{\pm8.1}$ | $64.69_{\pm9.6}$ | $60.94_{\pm8.5}$ |
| GIGAP [64] | ABMIL [25] | ✗ | $71.85_{\pm2.08}$ | $60.56_{\pm8.6}$ | $63.81_{\pm9.3}$ | $59.85_{\pm8.1}$ |
| | TransMIL [48] | | $65.45_{\pm2.04}$ | $60.40_{\pm8.8}$ | $62.90_{\pm9.2}$ | $60.12_{\pm8.5}$ |
| | GIGAP [64] | | $65.86_{\pm2.22}$ | $62.99_{\pm8.7}$ | $62.64_{\pm9.3}$ | $57.63_{\pm5.4}$ |
| | RRTMIL [56] | | $72.46_{\pm1.74}$ | $59.69_{\pm8.7}$ | $66.43_{\pm8.8}$ | $57.81_{\pm8.4}$ |
| | 2DMamba [67] | | $75.72_{\pm2.02}$ | $64.49_{\pm7.0}$ | $65.35_{\pm9.6}$ | $57.58_{\pm7.9}$ |
| ResNet-50 | Best-of-two-stage | ✗ | $62.72_{\pm2.15}$ | $64.15_{\pm8.1}$ | $64.93_{\pm9.1}$ | $60.78_{\pm8.2}$ |
| ResNet-18 | C2C [49] | ✓ | 62.91 +0.19 | - | - | - |
| ResNet-50 | FT [30] | ✓ | 66.06 +3.34 | - | - | - |
| ResNet-18 | ABMILX (ours) | ✓ | 78.34 +15.6 | 64.91 +0.76 | 67.78 +2.85 | 61.20 +0.42 |
| ResNet-50 | ABMILX (ours) | ✓ | **78.83 +16.1** | 64.72 +0.57 | 67.20 +2.27 | 60.78 +0.00 |

NSCLC. *We suggest that the two-stage method based on FMs has saturated performance on classical tasks and is bottlenecked by the lack of encoder adaptation in challenging tasks.*

**ABMILX Shows E2E Potential.** Through E2E learning with ABMILX and downstream data, we achieve FMs-level performance using ResNet models that were pre-trained in ImageNet-1k. It outperforms FMs on multiple challenging datasets (+4% Acc. on PANDA, +0.8% AUC on BRCA). Moreover, the E2E learning cost of ABMILX is substantially lower than the pretraining cost of FMs, approaching the cost of training second-stage aggregators, with more details provided in next section. Additionally, we show that fine-tuning upstream pre-trained aggregators, like CHIEF and GIGAP, did not yield the desired results. This further underscores the necessity of E2E training of encoders and aggregators with slide supervision. In particular, we demonstrate the scalability of the proposed method with respect to the model size. Except for survival analysis influenced by the sampling numbers (Table 2), R50 shows a general improvement compared to R18. Most critically, we validated the generalization ability through external validation on the CPTAC dataset (Table 3). A ResNet-50 encoder trained on TCGA using our E2E framework not only shows superior generalization but also outperforms UNI, a ViT-L pre-trained on over one billion pathology patches. This result validates that our E2E learning approach fosters robust transferability that can overcome cross-dataset domain shifts, rivaling the benefits of massive-scale pre-training. In conclusion, empowered by ABMILX, we present the impact and enormous potential of E2E learning in CPath. We also present more discussion in Appendix C.1.

Table 3: Performance of different methods on external validation from TCGA to CPTAC datasets.

| Encoder | Method | E2E | CPTAC-NSCLC (AUC↑) | CPTAC-LUAD (C-index↑) |
|---|---|---|---|---|
| ResNet-50 | ABMIL [25] | ✗ | 66.42 | 46.34 |
| | TransMIL [48] | ✗ | 74.59 | 48.24 |
| | WIKG [33] | ✗ | 64.04 | OOM |
| UNI [11] | ABMIL [25] | ✗ | 83.73 | 53.59 |
| | TransMIL [48] | ✗ | **85.24** | 51.36 |
| | WIKG [33] | ✗ | 83.56 | OOM |
| ResNet-50 | ABMILX (ours) | ✓ | 85.19 | **54.00** |

Table 4: **Top:** Comparison of computational cost. TTime (3090 GPU hours) denotes Train Time on BRCA-subtyping. We add the features extraction time for two-stage methods. IT (s / slide) is the Inference Time. Memory (GB) is the GPU memory evaluated with batch size 1 and fp16 during training. **Bottom:** Abalation of ABMILX in E2E training.

| Encoder | Method | E2E | Pretrain Cost↓ | TTime↓ | Memory↓ | IT↓ | Grad.↑ | Sub.↑ | Surv.↑ |
|---|---|---|---|---|---|---|---|---|---|
| CHIEF [62] | ABMIL [25] | ✗ | 32GB× - h | 1+2h | 2G | 6.2s | 65.66 | 91.09 | 64.02 |
| UNI [11] | TransMIL [48] | ✗ | 80GB× - h | 1+7h | 8G | 25s | 68.06 | 93.33 | 60.45 |
| GIGAP [64] | GIGAP [64] | ✗ | 80GB×3,072h | 6+23h | 7G | 83s | 65.86 | 93.72 | 62.64 |
| ResNet-18 | ABMILX + MRIS (ours) | ✓ | - | 9h | 9G | **1.7s** | 78.34 | 93.97 | 67.78 |

(a) Survival  (b) Grading  (c) Sub-typing

## 4.3 Ablation Study

In this subsection, we systematically investigate the impact of MIL in E2E training and ablate the ABMILX. Unless otherwise specified, all ablation experiments use ResNet-18 as the encoder. For the survival analysis task, we utilize the larger BRCA dataset. All efficiency experiments are conducted on the BRCA-subtyping benchmark. To evaluate model inference speed, we use an input size of $1\times10000\times3\times224\times224$, representing the average data volume processed in clinical scenarios.

**MIL Matters in End-to-End Trainings.** Right Table shows the impact of the sampling and aggregation modules on E2E learning. We observe that different MILs have a significant effect on E2E learning performance. Specifically, ABMIL exhibits unsatisfactory performance in E2E training, except on the PANDA dataset. We attribute this to its excessive sparsity hindering E2E optimization. The PANDA dataset contains fewer patches

| Method | TTime↓ | Grad.↑ | Sub.↑ | Surv.↑ |
|---|---|---|---|---|
| *Different MIL Models with MRIS* | | | | |
| ABMIL [25] | 9h | 75.46 | 89.23 | 62.70 |
| DSMIL [29] | 9h | 76.28 | 91.09 | 64.32 |
| TransMIL [48] | 10h | 75.08 | 91.44 | 63.42 |
| RRTMIL(AB.) [56] | 9h | 17.99 | 61.82 | 53.42 |
| ABMILX | 9h | **78.34** | **93.97** | **67.78** |
| *Different Sampling Strategies with ABMILX* | | | | |
| Attention Sampling | 68h | 77.43 | 93.14 | 66.53 |
| Random Sampling | 9h | 76.77 | 92.72 | 67.24 |

per slide (500 vs. 10,000 for TCGA-BRCA), enabling MIL to focus on discriminative regions more easily, thus suffering minimal impact on E2E optimization. RRTMIL exacerbates this problem, leading to optimization collapse. This complex MIL method, with a serial feature re-embedding module preceding ABMIL, makes E2E training more fragile. It further impairs the representation of features already affected by sparse attention, accelerating the collapse of the optimization loop. TransMIL and DSMIL, the transformer-based methods, partially mitigate this issue. However, relying solely on global attention struggles to focus on key regions within the numerous redundant patches in training, resulting in a considerable performance gap compared to FMs. ABMILX, while maintaining desirable sparsity, alleviates optimization issues and achieves significant performance improvements. Furthermore, complex sampling strategies, such as attention-based sampling, offer only limited performance gains compared to vanilla random sampling. Such strategies require patch evaluation and incur substantial training time (TTime). Multi-scale random instance sampling (MRIS) shows better performance. Appendix C.2 provide further discussion.

**Validity of Our ABMILX.** Table 4 (bottom) ablates key components of ABMILX. E2E training with ABMIL performs poorly except for PANDA due to optimization challenges. It performs below SOTA MIL with R50 features and significantly underperforming FMs. After introducing multi-head mechanisms, the extreme focus on redundant instances caused by sparse attention is effectively mitigated, thus achieving consistent improvements. More importantly, by refining attention using global patch correlations in the attention plus module, optimization issues are further alleviated. This improvement helps ABMILX achieve FMs-level performance. Furthermore, the sharp performance degradation when freezing the encoder demonstrates the necessity of E2E learning. We also validate ABMILX under two-stage paradigm in Appendix C.3.

**Computational Cost Analysis.** Table 4 (top) shows that the significant computational cost of FMs is attributed to pre-training and inference. The resource consumption of FM pre-training increases rapidly with model size. Large models also severely impact their clinical application, with FMs taking up to 83 seconds to process a single slide, excluding data pre-processing. Although feature input reduces the cost of the second-stage training, increasingly complex aggregators continue to increase training time and memory consumption. In contrast, our E2E training pipeline maintains a lower computational cost. Specifically, we do not require additional pre-training, and the overall training time and memory consumption are comparable to traditional second-stage feature-based training. Benefiting from the effectiveness of E2E learning, our pipeline offers significant advantages for clinical applications. It achieves competitive performance with only 1/50 of the inference time.

### 4.4 Qualitative Results

**Feature Visualization.** To validate that the performance gains of E2E training stem from task-specific encoder fine-tuning, we visualize instance features from the PANDA dataset using UMAP [22] in right figure. Features extracted offline by a ResNet pre-trained on ImageNet exhibit a dispersed distribution in the feature space, with poor separation between tumor and normal instances. Pre-training helps UNI provide a preliminary separation of instance types, but instances with the same annotations are not densely clustered. In contrast, after E2E learning with our proposed ABMILX, the ResNet-extracted features demonstrate improved inter-class separability and intra-class compactness.

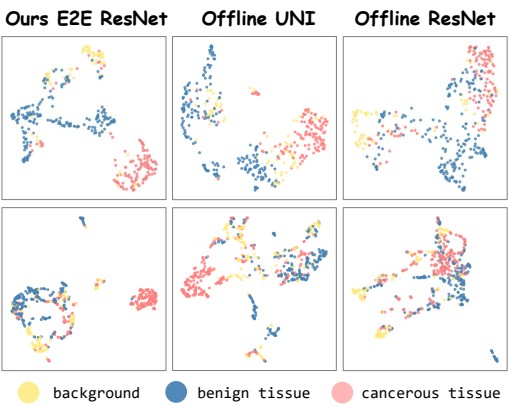

## 5 Conclusion

The lack of well-adapted offline features and disjointly optimized models has become a performance bottleneck in CPath. While slide-level supervised E2E learning presents a fundamental solution, it remains underexplored due to efficiency and performance challenges. Our work revisits slide-level supervised E2E learning in CPath from the MIL perspective. We demonstrate the impact of sparse-attention MIL on E2E optimization. After addressing optimization challenges through the proposed ABMILX, we show that E2E-trained ResNet achieves comparable performance to foundation models with lower computational costs. We believe E2E learning has the potential to benefit upstream pre-training and achieve further breakthroughs with increased computational resources. Revisiting the role of MIL in E2E learning may be key to realizing its potential.

## Acknowledgements

This work was supported by Shenzhen Science and Technology Program (JCYJ20240813114237048), "Science and Technology Yongjiang 2035" key technology breakthrough plan project (2024Z120), Chinese government-guided local science and technology development fund projects (scientific and technological achievement transfer and transformation projects) (254Z0102G), Supercomputing Center of Nankai University (NKSC).

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

# Appendix

## Table of Contents

# A  Theoretical Analysis of Optimization Risk in End-to-End Training

## A.1  Definition of Optimization Risk

**Definition 1.** (Optimization Risk) Let $\mathcal{N}$ denote the set of noisy instances, $\mathcal{D}$ denote discriminative instances, and $O(\hat{a})$ is a measure of the contribution of attention $\hat{a}$ to the final bag feature $Z$. $O(\cdot)$ is monotone increasing. We define the optimization risk $\mathcal{R}$ as the impact of maximum attention value among noisy instances to bag feature:

$$\mathcal{R} = O(\max_{i \in \mathcal{N}} \hat{a}_i), \quad \text{where } \hat{a}_i = \frac{\exp(a_i)}{\sum_{k=1}^{s} \exp(a_k)}, \tag{6}$$

where $s$ is the number of sampled instances, and $a_i$ is derived from the instance feature $e_i$ after MLP, $a_i = \text{MLP}(e_i)$, $A = \{a_1, ..., a_i, ...a_s\}$.

**Rationale for Max Operator in Optimization Risk.** The use of the maximum operator in Definition 1, as opposed to a summation, is crucial for accurately modeling the optimization risk. Under effective learning conditions, the model is expected to assign significantly higher attention to discriminative instances ($\mathcal{D}$) compared to noisy instances ($\mathcal{N}$) with high probability. Let us assume that for the set of noisy instances $\mathcal{N}$, there exists an attention value $\hat{a}_{\mathcal{N}}$ such that the summed contribution of noisy instances can be approximated by $O(\hat{a}_{\mathcal{N}}) \cdot |\mathcal{N}| \approx \sum_{i \in \mathcal{N}} O(\hat{a}_i)$. Similarly, let $\hat{a}_{\mathcal{D}}$ represent a typical attention value for a discriminative instance. In well-behaved scenarios, $\hat{a}_{\mathcal{D}} > \hat{a}_{\mathcal{N}}$, implying that the collective contribution of noisy instances to the bag feature $Z$ is relatively small, and thus the optimization risk is low.

However, the primary concern for optimization risk arises when, even with low probability, the MIL model assigns an a typically large attention value to a single noisy instance. If this single noisy instance significantly influences $Z$, the backpropagation process can adversely affect the optimization of the feature encoder, leading to suboptimal instance features $e_i$. This, in turn, can result in poorer attention scores $a_i$, creating a detrimental feedback loop. The max operator specifically targets this scenario by focusing on the worst-case contribution from a single noisy instance. A larger variance in attention values within $\mathcal{N}$, particularly the presence of outliers with high attention, directly translates to a higher optimization risk as measured by the max operator.

*ABMILX is designed to mitigate this optimization risk by intervening when noisy instances receive unduly high attention, thereby preventing their disproportionate impact $O(\max_{i \in \mathcal{N}} \hat{a}_i)$, and breaking the aforementioned vicious cycle.*

## A.2  Multi-head Local Attention Mechanism

For ABMILX with $m$ heads, each head generates independent attention distributions $\hat{A}_{\text{MHLA}} = \{\hat{A}^{(1)}, ..., \hat{A}^{(j)}, ..., \hat{A}^{(m)}\}$ through distinct feature subspaces. The correlation of different heads can be defined by $\text{Corr}(\hat{A}^{(j)}, \hat{A}^{(k)}) \in [0, 1]$. We can obtain the following derivation:

$$\underbrace{\text{Corr}\left(\hat{a}_i^{(j)}, \hat{a}_i^{(k)}\right) \to 1, \forall i, j, k}_{\text{multi-head attention collapses to full correlation}} \Rightarrow \mathcal{R}_{\text{MHLA}} = \mathcal{R}_{\text{ABMIL}} \tag{7}$$

$$\underbrace{\text{Corr}\left(\hat{a}_i^{(j)}, \hat{a}_i^{(k)}\right) \to 0, \ \forall i, j, k}_{\text{multi-head attention is fully independent}} \tag{8}$$

$$\Rightarrow \mathcal{R}_{\text{MHLA}} = \sum_{j}^{m} \mathcal{R}^{(j)} < \underbrace{m \cdot \max_{1 \le j \le m} \mathcal{R}^{(j)}}_{\text{upper bound of multi-head risk}} = m \cdot \underbrace{\frac{1}{m} \mathcal{R}_{\text{ABMIL}}}_{\text{each head have 1/m impact}} \tag{9}$$

Under the diversity assumption of head specialization, the optimization risk satisfies:

$$\mathcal{R}_{\text{MHLA}} < \mathcal{R}_{\text{ABMIL}} \tag{10}$$

The Eq.(9) can be proved by:

$$Z^{(j)} = \sum_i \hat{a}_i^{(j)} e_i^{(j)}, \;\; Z = \left\{ Z^{(1)}, ..., Z^{(m)} \right\}, \;\; Z \in \mathbb{R}^{1 \times D}, \;\; Z^{(j)} \in \mathbb{R}^{1 \times \lceil D/m \rceil} \tag{11}$$

$$\Rightarrow \max_{1 \leq j \leq m} \mathcal{R}^{(j)} = \max_{1 \leq j \leq m} \frac{1}{m} O(\max_{i \in \mathcal{N}} \hat{a}_i^{(j)}) = \frac{1}{m} \mathcal{R}_{\text{ABMIL}} \tag{12}$$

the risk of each head only affects the $\frac{1}{m}$ dimension in $Z$, $O_{\text{mul-head}}^{(j)} = \frac{1}{m} O$.

## A.3 Attention Plus Propagation

In order to simplify the analysis, this subsection does not deal with the multi-head mechanisms. Let $U$ be the normalized feature similarity matrix where $U_{ij} = \text{sim}(e_i, e_j) \in [0, 1]$. The attention refinement:

$$\tilde{a}_i = a_i + \alpha \sum_{k=1}^{s} U_{ik} a_k. \tag{13}$$

For noisy instance with highest original attention $\hat{a}' = \max_{i \in \mathcal{N}} \hat{a}_i, \;\; i' = \arg\max_{i \in \mathcal{N}} \hat{a}_i$, when the propagation term between discriminative instances $\sum_{n \in \mathcal{D}} \sum_{k \in \mathcal{D}} U_{nk} a_k$ are higher that between the noise ones $\sum_{j \in \mathcal{N}} \sum_{k \in \mathcal{N}} U_{jk} a_k$, the post-softmax effect yields:

$$\frac{\exp(\tilde{a}')}{\sum_{k=1}^{s} \exp(\tilde{a}_k)} \leq \frac{\exp(a')}{\sum_{k=1}^{s} \exp(a_k)} \tag{14}$$

$$O\left(\frac{\exp(\tilde{a}')}{\sum_{k=1}^{s} \exp(\tilde{a}_k)}\right) \leq O\left(\frac{\exp(a')}{\sum_{k=1}^{s} \exp(a_k)}\right) \Rightarrow \mathcal{R}_{\text{A+}} \leq \mathcal{R}_{\text{ABMIL}} \tag{15}$$

(**Proof of Eq.**(14)) Substituting Eq.(13) into the softmax operation:

$$\frac{\exp(\tilde{a}')}{\sum_{k=1}^{s} \exp(\tilde{a}_k)} = \frac{\exp\left(a' + \alpha \sum_n U_{i'n} a_n\right)}{\sum_{k=1}^{s} \exp\left(a_k + \alpha \sum_n U_{kn} a_n\right)} \tag{16}$$

Let $\Delta_k = \alpha \sum_n U_{kn} a_n$ represent the attention modulation term. We decompose the attention:

$$\frac{\exp(a' + \Delta')}{\sum_{k=1}^{s} \exp(a_k + \Delta_k)} = \underbrace{\frac{\exp(a')}{\sum_{k=1}^{s} \exp(a_k)}}_{\text{Original risk}} \times \underbrace{\frac{\exp(\Delta')}{\sum_{k=1}^{s} \exp(\Delta_k) \hat{a_k}}}_{\text{Modulation factor } \Lambda} \tag{17}$$

The critical inequality $\Lambda \leq 1$ holds when:

$$\exp(\Delta') \leq \sum_{k=1}^{s} \exp(\Delta_k) \hat{a}_k \tag{18}$$

By Jensen's inequality:

$$\sum_{k}^{s} \hat{a}_k \exp(\Delta_k) \geq \exp\left(\sum_{k}^{s} \hat{a}_k \Delta_k\right) \tag{19}$$

$$\sum_{k}^{s} \hat{a}_k \Delta_k = \alpha \sum_{n=1}^{s} \hat{a}_n \sum_{k=1}^{s} U_{nk} a_k, \Delta_j = \alpha \sum_{n=1}^{s} \hat{a}_m \sum_{k=1}^{s} U_{jk} a_k \quad (j \in \mathcal{N}) \tag{20}$$

Through the subtraction operation $\sum_k^s \hat{a}_k \Delta_k - \Delta_j$, we derive:

$$\sum_k^s \hat{a}_k \Delta_k - \Delta_j = \alpha \sum_{n \in \mathcal{D}} \hat{a}_n \sum_{k=1}^s (\boldsymbol{U}_{nk} - \boldsymbol{U}_{jk}) a_k + \alpha \underbrace{\sum_{n \in \mathcal{N}} \hat{a}_n \sum_{k=1}^s (\boldsymbol{U}_{nk} - \boldsymbol{U}_{jk}) a_k}_{\text{Intra-class Modulation Difference } \mathcal{Q}} \tag{21}$$

$$= \alpha \sum_{n \in \mathcal{D}} \hat{a}_n \sum_{k \in \mathcal{D}} \boldsymbol{U}_{nk} a_k - \alpha \sum_{n \in \mathcal{D}} \hat{a}_n \sum_{k \in \mathcal{N}} \boldsymbol{U}_{jk} a_k + \mathcal{Q} + \underbrace{\alpha \sum_{n \in \mathcal{D}} \hat{a}_n \sum_{k \in \mathcal{N}} \boldsymbol{U}_{nk} a_k - \alpha \sum_{n \in \mathcal{D}} \hat{a}_n \sum_{k \in \mathcal{D}} \boldsymbol{U}_{jk} a_k}_{\text{Cross-class Propagation } \mathcal{T}}$$

$$\tag{22}$$

$$= \alpha \sum_{n \in \mathcal{D}} \hat{a}_n \sum_{k \in \mathcal{D}} \boldsymbol{U}_{nk} a_k - \alpha \sum_{n \in \mathcal{D}} \hat{a}_n \sum_{k \in \mathcal{N}} \boldsymbol{U}_{jk} a_k + \mathcal{Q} + \mathcal{T} \tag{23}$$

$$= \alpha \sum_{n \in \mathcal{D}} \hat{a}_n \left( \sum_{k \in \mathcal{D}} \boldsymbol{U}_{nk} a_k - \sum_{k \in \mathcal{N}} \boldsymbol{U}_{jk} a_k \right) + \mathcal{Q} + \mathcal{T} \tag{24}$$

As discriminative instances $i \in \mathcal{D}$ and noise ones $j \in \mathcal{N}$ typically share weak cross-class correlations $\boldsymbol{U}_{ij}/\boldsymbol{U}_{ji}$ with each other but have stronger correlations within classes, the values of $\mathcal{Q}$ and $\mathcal{T}$ could be ignored, $\mathcal{Q}, \mathcal{T} \ll \alpha \sum_{n \in \mathcal{D}} \hat{a}_n \sum_{k \in \mathcal{D}} \boldsymbol{U}_{nk} a_k, \alpha \sum_{n \in \mathcal{D}} \hat{a}_n \sum_{k \in \mathcal{N}} \boldsymbol{U}_{jk} a_k$. As mentioned before, $\boldsymbol{A}$ is a sparse attention vector. Therefore, $\Lambda \leq 1$ holds when the number of high value instances in $\mathcal{D}$ is more than that in $\mathcal{N}$:

$$\sum_{k=1}^s \hat{a}_k \Delta_k - \Delta_j \geq 0 \tag{25}$$

$$\sum_{k=1}^s \hat{a}_k \Delta_k \geq \Delta_j \tag{26}$$

$$\sum_{k=1}^s \hat{a}_k \exp(\Delta_k) \geq \exp\left( \sum_{k=1}^s \hat{a}_k \Delta_k \right) \geq \exp(\Delta') \tag{27}$$

In this case, the highest attention $\hat{a}'$ will be suppressed, yielding lower risks and higher benefits:

$$\frac{\exp(\tilde{a}')}{\sum_{k=1}^s \exp(\tilde{a}_k)} = \frac{\exp(a')}{\sum_{k=1}^s \exp(a_k)} \times \Lambda \leq \hat{a}' \tag{28}$$

$$\mathcal{R}_{\text{A+}} = O\left( \frac{\exp(\tilde{a}')}{\sum_{k=1}^s \exp(\tilde{a}_k)} \right) = O\left( \frac{\exp(a')}{\sum_{k=1}^s \exp(a_k)} \times \Lambda \right) \leq \mathcal{R}_{\text{ABMIL}} \tag{29}$$

**(Proof of Eq.(17))**

$$\sum_{k=1}^s \exp(a_k + \Delta_k) = \sum_{k=1}^s \exp(a_k) \exp(\Delta_k) \tag{30}$$

With post-softmax attention definition, $\hat{a}_k = \frac{\exp(a_k)}{\sum_{n=1}^s \exp(a_n)}$, we can obtain:

$$\exp(a_k) = \hat{a}_k \times \sum_{n=1}^s \exp(a_n) \tag{31}$$

Substituting Eq.(31) into Eq.(30):

$$\sum_{k=1}^s \exp(a_k + \Delta_k) = \sum_{k=1}^s \left( \hat{a}_k \times \sum_{n=1}^s \exp(a_n) \right) \exp(\Delta_k)$$
$$= \left( \sum_{n=1}^s \exp(a_n) \right) \times \sum_{k=1}^s \hat{a}_k \exp(\Delta_k) \tag{32}$$

Thus, the original softmax expression can be decomposed as:

$$\frac{\exp(a' + \Delta')}{\sum_{k=1}^s \exp(a_k + \Delta_k)} = \frac{\exp(a')}{\sum_{n=1}^s \exp(a_n)} \times \frac{\exp(\Delta')}{\sum_{k=1}^s \hat{a}_k \exp(\Delta_k)} \tag{33}$$

## A.4 Empirical Validation of Theoretical Analysis

Our theoretical analysis posits that the maximum attention score of noisy instances serves as a proxy for optimization risk. The theory further suggests that ABMILX can mitigate this risk while maintaining reasonable sparsity. To empirically validate this connection, we introduce the MAX-N metric, defined as the product of the maximum attention score of noisy instances for each slide and the total number of instances.

We measured MAX-N on the CAMELYON dataset during the early E2E training stage, with the results reported in Table 5. The experimental results align closely with our theoretical analysis. ABMILX drastically reduces the MAX-N score (our risk proxy) from 21.2162 (for ABMIL) to 2.6557, demonstrating its effectiveness in mitigating optimization risk. Concurrently, it maintains a functional sparsity level (36) compared to the baseline's (80), achieving the "reasonable sparsity" predicted by our theory. This effective risk mitigation and balanced sparsity directly translate to a substantial performance improvement (95.88% vs. 91.78%). These findings validate our theoretical framework, demonstrating a clear alignment between our analysis and the experimental results.

| Metric/Method | ABMIL | ABMILX |
|---|---|---|
| MAX-N | 21.2162 | 2.6557 |
| Sparsity | 80 | 36 |
| Performance | 91.78 | 95.88 |

Table 5: Empirical validation of ABMILX's risk mitigation on the CAMELYON dataset. ABMILX significantly reduces the optimization risk proxy (MAX-N) and balances sparsity, aligning with our theoretical analysis and leading to superior performance over the ABMIL baseline.

# B Datasets and Implementation Details

## B.1 Datasets

We validate our E2E training ABMILX on various computational pathology tasks, including cancer grading (PANDA [6]), subtyping (TCGA-NSCLC, TCGA-BRCA), survival analysis (TCGA-LUAD, TCGA-LUSC, TCGA-BLCA), and diagnosis (CAMELYON [3, 2]).

**PANDA** [6] (CC-BY-4.0) is a large-scale, multi-center dataset dedicated to prostate cancer detection and grading. It comprises 10,202 digitized H&E-stained whole-slide images, making it one of the most extensive public resources for prostate cancer histopathology. Each slide is annotated according to the Gleason grading system and subsequently assigned an International Society of Urological Pathology (ISUP) grade, enabling both cancer detection and severity assessment. The dataset includes a diverse distribution of ISUP grades, with 2,724 slides classified as grade 0 (benign), 2,602 as grade 1, 1,321 as grade 2, 1,205 as grade 3, 1,187 as grade 4, and 1,163 as grade 5. Spanning multiple clinical centers, PANDA ensures a broad range of samples, mitigating center-specific biases.

The Non-Small Cell Lung Cancer (**NSCLC**) project of The Cancer Genome Atlas (TCGA) by the National Cancer Institute is the primary dataset for the cancer sub-typing task. **TCGA-NSCLC** is the most common type of lung cancer, accounting for approximately 85% of all lung cancer cases. This classification includes several subtypes, primarily Lung Adenocarcinoma (**LUAD**) and Lung Squamous Cell Carcinoma (**LUSC**). The dataset contains 541 slides from 478 LUAD cases and 512 slides from 478 LUSC cases, with only image-level labels provided.

The Breast Invasive Carcinoma (**TCGA-BRCA**) project is another sub-typing dataset we used. TCGA-BRCA includes two subtypes: Invasive Ductal Carcinoma (**IDC**) and Invasive Lobular Carcinoma (**ILC**). It contains 787 IDC slides and 198 ILC slides from 985 cases. To mitigate the impact of class imbalance on E2E optimization, we employed oversampling of the ILC class, resulting in a training set with an IDC:ILC ratio of 2:1.

The primary goal of survival analysis is to estimate the survival probability or survival time of patients over a specific period. Therefore, we used the **TCGA-LUAD**, **TCGA-BRCA**, and **TCGA-BLCA** projects to evaluate the model performance for survival analysis tasks. Unlike the diagnosis and

Table 6: Comparison between E2E methods and two-stage methods. We add the features extraction time on two-stage methods. *FM denotes the best performance achieved among foundation models (CHIEF [62], UNI [11], GIGAP [64]), with only the highest value reported. Existing E2E methods devote excessive resources to sampling, incurring long training times yet offering only marginal gains over two-stage ResNet50, and incorporating FM features further diminishes E2E's advantage. In contrast, our ABMILX drastically shortens training while improving performance, remaining competitive even against latest two-stage methods using FM features.

| Encoder | Method | TTime | Grad. | Sub. | Surv. |
|---|---|---|---|---|---|
| *Latest Two-stage Methods* | | | | | |
| R50 | WIKG [33] | 3h | 62.72 | 88.37 | 60.65 |
| | RRT [56] | 3h | 60.42 | 89.35 | 63.03 |
| FM* | WIKG [33] | 24h | 74.97 | 94.76 | 66.97 |
| | RRT [56] | 24h | 74.00 | 94.84 | 67.30 |
| *E2E Training* | | | | | |
| R18 | C2C [49] | 84h | 62.91 | 91.13 | - |
| R50 | FT [61] | 45h | 66.06 | 86.48 | - |
| R18 | ABMILX | 9h | 78.34 | 93.97 | **67.78** |
| R50 | ABMILX | 22h | **78.83** | **95.17** | 67.20 |

sub-typing tasks, the survival analysis datasets are case-based rather than WSI-based. The WSIs of TCGA-LUAD and TCGA-BRCA are identical to those used in the sub-typing task but with different annotations. The TCGA-BLCA dataset includes 376 cases of bladder urothelial carcinoma.

We supplemented the **CAMELYON** dataset (CC-BY-4.0) to evaluate qualitative and quantitative results of different methods. The dataset comprises CAMELYON-16 [3] and CAMELYON-17 [2], which are among the largest publicly available datasets for breast cancer lymph node metastasis diagnosis, each providing binary labels (metastasis or not). The CAMELYON dataset contains 899 WSIs (591 negative and 308 positive) from 370 cases. Additionally, 159 slides have complete pixel-level annotations, making this dataset particularly suitable for qualitative analysis.

We randomly split the PANDA dataset into training, validation, and testing sets with a ratio of 7:1:2. Due to the limited data size, the remaining datasets are divided into training and testing sets with a ratio of 7:3.

## B.2    Preprocess

**End-to-End Training.** To efficiently process gigapixel slides in our E2E pipeline, we first crop each WSI into a series of non-overlapping patches of size (256 × 256) and discard background regions, including holes, as in CLAM [38]. Except for PANDA, we perform patching at 10x magnification, with an average of approximately 3,000 patches per slide. For the PANDA dataset, we process patches at 40x magnification, resulting in an average of 505 patches. Additionally, we extract patches at 5x and 20x magnifications to support our multi-scale random sampling strategy. After patch extraction, we store all data using the LMDB (Lightning Memory-Mapped Database) format. Ultimately, each dataset contains an average of 4~6 million 256×256 patches.

**Two-Stage Framework.** Following prior works [38, 48, 66, 56], we crop each WSI into a series of non-overlapping patches of size (256 × 256) at 20× magnification and discard the background regions, including holes, as in CLAM [38]. The average number of patches per dataset is around 10,000. To efficiently handle the large number of patches, we follow the traditional two-stage paradigm, using a pre-trained offline model to extract patch features. This includes a ResNet-50 [21] pre-trained on ImageNet-1k [16]. Specifically, the last convolutional module of the ResNet-50 is removed, and a global average pooling is applied to the final feature maps to generate the initial feature vector. Additionally, we also use state-of-the-art foundation models pre-trained on WSIs, such as CHIEF [62], GigaPath [64] and UNI [11].

Table 7: More MIL aggregators in E2E training. We categorize these aggregators to two type: Sparse-Attention (S.A.) and Transformer-like (Trans.). We categorize these aggregators into two types: Sparse-Attention (S.A.) and Transformer-like (Trans.). Existing S.A. methods, primarily focused on the two-stage paradigm, face challenges in E2E optimization. While Transformer-based methods partially alleviate the optimization challenges caused by extreme sparsity, they struggle to focus on key regions within the numerous redundant patches in the E2E training. It leads to a noticeable performance gap compared to two-stage methods under FM features (marked in gray).

| Aggregator | Aggr. Type | Grad. | Sub. | Surv. |
|---|---|---|---|---|
| Best in FMs | - | 74.97 | 94.84 | 67.30 |
| ABMIL [25] | S.A. | 75.46 | 89.23 | 62.70 |
| RRTMIL [56] | S.A. | 17.99 | 61.82 | 53.42 |
| QAMIL | Trans. | 75.12 | 90.65 | 64.29 |
| TransMIL [48] | Trans. | 75.08 | 91.44 | 63.42 |
| DSMIL [29] | Trans. | 76.28 | 91.09 | 64.32 |
| VITMIL | Trans. | 76.98 | 92.61 | 63.67 |
| ABMILX | S.A. | 78.34 | 93.97 | 67.78 |

## B.3 Implementation Details

**End-to-End Training.** To maintain consistency with traditional ResNet-based two-stage methods, we removed the last stage module of ResNet. For MIL, we added a LayerNorm [1] layer at its input to better optimize the Encoder. We also disabled the bias of all fully connected layers in MIL, which we found beneficial for E2E optimization. For training, we used different hyperparameters for different tasks. For cancer grading (PANDA), we employed an Adam [28] optimizer with a learning rate of $2 \times 10^{-4}$ and a weight decay of $1 \times 10^{-5}$, training for 200 epochs. For sub-typing (NSCLC, BRCA), we used an AdamW [37] optimizer with a learning rate of $8 \times 10^{-5}$ and no weight decay, training for 75 epochs. For survival analysis (LUAD, BLCA, BRCA), we utilized an AdamW optimizer with a learning rate of $8 \times 10^{-5}$ and a weight decay of $5 \times 10^{-2}$, training for 30 epochs. The learning rate was adjusted using the Cosine annealing strategy. During training, we applied simple geometric data augmentations such as flipping and RandomResizedCrop. Since slides are typically H&E-stained, we found that color-related data augmentations could significantly impact performance. All experiments are conducted on 3090 GPUs. We adjusted the batch size based on the 24GB memory limit and the number of samples in different datasets. Unless otherwise specified, all ablation experiments use ResNet-18 as the encoder. All efficiency experiments are conducted on the BRCA-subtyping benchmark. We calculate the FLOPs for all models using an input size of $1 \times 512 \times 3 \times 224 \times 224$, which simulates the sampling of 512 patches during E2E training. To evaluate model inference speed, we use an input size of $1 \times 10000 \times 3 \times 224 \times 224$, representing the average data volume processed in clinical scenarios.

**Two-Stage Framework.** Following [38, 48, 56], the offline feature is projected to a 512-dimensional feature vector using a fully-connected layer. For features extracted by ResNet-50, an AdamW optimizer [37] with a learning rate of $2 \times 10^{-4}$ and no weight decay is used for model training. For features extracted by foundation models, the learning rate is changed to $1 \times 10^{-4}$. The learning rate is adjusted using the Cosine annealing strategy. All models are trained for 200 epochs for cancer grading. For sub-typing and survival analysis tasks, the number of epochs is reduced to 75. Notably, due to the training of the GIGAP [64] aggregator exceeding the memory limit of a 3090 GPU, we sampled the number of patches to 1024 during its training. For all two-stage methods except GIGAP aggregator, following [38, 48, 56], we used the complete patch sequence for training. Due to the variable sequence length, we conventionally set the batch size to 1. We employed unified hyperparameters to train all methods.

Table 8: CPath performance of different methods. Our methods are marked in gray . OOM denotes Out-of-Memory. Two-stage paradigms based on FMs have achieved saturated performance on classical tasks (CAMELYON and NSCLC) leveraging advantages of over 100K pre-trained pathology slides. However, these approaches are bottlenecked by the lack of encoder adaptation in challenging benchmarks (BRCA-subtyping and Survival Analysis). We further show the results of ABMILX in two-stage paradigms, demonstrating its versatility as a pure MIL architecture in CPath tasks without requiring hyperparameter tuning.

| Encoder | Aggregator | Diagnosis | Grading | Sub-typing | | Survival Analysis | | |
|---|---|---|---|---|---|---|---|---|
| | | CAMELYON | PANDA | BRCA | NSCLC | LUAD | BRCA | BLCA |
| R50 [21] (26M Para.) (ImageNet-1K) | ABMIL [25] | $91.84_{\pm 4.0}$ | $58.89_{\pm 0.8}$ | $83.80_{\pm 6.6}$ | $92.32_{\pm 2.7}$ | $59.56_{\pm 8.6}$ | $64.93_{\pm 9.1}$ | $55.01_{\pm 7.9}$ |
| | CLAM [38] | $91.85_{\pm 4.0}$ | $59.45_{\pm 2.2}$ | $85.86_{\pm 6.4}$ | $92.28_{\pm 2.7}$ | $59.79_{\pm 8.7}$ | $62.90_{\pm 9.4}$ | $55.78_{\pm 8.0}$ |
| | TransMIL [48] | $90.59_{\pm 4.2}$ | $56.42_{\pm 2.1}$ | $88.52_{\pm 5.4}$ | $92.49_{\pm 2.7}$ | $64.15_{\pm 8.1}$ | $59.15_{\pm 10.1}$ | $56.96_{\pm 8.4}$ |
| | DSMIL [29] | $92.63_{\pm 3.5}$ | $61.24_{\pm 2.3}$ | $85.68_{\pm 6.1}$ | $91.12_{\pm 3.0}$ | $61.70_{\pm 8.6}$ | $61.96_{\pm 9.5}$ | $56.22_{\pm 8.2}$ |
| | DTFD [66] | $91.88_{\pm 4.0}$ | $60.62_{\pm 1.1}$ | $83.46_{\pm 7.2}$ | $92.36_{\pm 2.7}$ | $59.47_{\pm 8.5}$ | $61.76_{\pm 10.3}$ | $58.60_{\pm 8.2}$ |
| | WIKG [33] | $91.42_{\pm 4.0}$ | $62.72_{\pm 2.2}$ | $88.37_{\pm 5.4}$ | $92.57_{\pm 2.5}$ | OOM | $60.65_{\pm 9.2}$ | OOM |
| | RRTMIL [56] | $94.19_{\pm 3.2}$ | $61.97_{\pm 2.2}$ | $89.35_{\pm 5.4}$ | $94.43_{\pm 2.2}$ | $62.19_{\pm 8.4}$ | $63.03_{\pm 10.2}$ | $60.78_{\pm 8.2}$ |
| | ABMILX | $92.37_{\pm 3.7}$ | $61.05_{\pm 2.2}$ | $87.45_{\pm 5.8}$ | $93.28_{\pm 2.4}$ | $61.25_{\pm 8.5}$ | $63.79_{\pm 9.4}$ | $58.64_{\pm 8.5}$ |
| CHIEF [62] (27M Para.) (Slide-60K) | ABMIL | $90.04_{\pm 6.0}$ | $65.66_{\pm 2.1}$ | $91.09_{\pm 4.7}$ | $96.22_{\pm 1.7}$ | $62.09_{\pm 8.8}$ | $64.02_{\pm 9.0}$ | $60.78_{\pm 8.6}$ |
| | TransMIL | $95.30_{\pm 3.8}$ | $60.89_{\pm 2.2}$ | $91.41_{\pm 4.0}$ | $96.39_{\pm 1.7}$ | $65.55_{\pm 8.3}$ | $61.46_{\pm 9.4}$ | $58.83_{\pm 8.3}$ |
| | CHIEF | $89.61_{\pm 6.2}$ | $64.24_{\pm 2.1}$ | $91.43_{\pm 4.5}$ | $96.84_{\pm 1.5}$ | $60.29_{\pm 8.1}$ | $67.95_{\pm 8.5}$ | $59.63_{\pm 8.3}$ |
| | ABMILX | $91.59_{\pm 4.7}$ | $65.82_{\pm 2.2}$ | $92.30_{\pm 4.2}$ | $96.60_{\pm 1.5}$ | $62.71_{\pm 8.7}$ | $66.08_{\pm 9.6}$ | $60.09_{\pm 8.2}$ |
| UNI [11] (307M Para.) (Mass-100K) | ABMIL | $96.58_{\pm 3.0}$ | $74.69_{\pm 2.1}$ | $94.05_{\pm 3.5}$ | $97.04_{\pm 1.6}$ | $59.65_{\pm 8.6}$ | $67.05_{\pm 10.2}$ | $57.29_{\pm 8.6}$ |
| | TransMIL | $96.63_{\pm 2.8}$ | $68.06_{\pm 2.1}$ | $93.33_{\pm 3.5}$ | $97.27_{\pm 1.6}$ | $60.43_{\pm 9.4}$ | $62.76_{\pm 10.5}$ | $60.45_{\pm 8.6}$ |
| | ABMILX | $96.77_{\pm 3.1}$ | $78.54_{\pm 1.9}$ | $93.34_{\pm 3.9}$ | $97.81_{\pm 1.2}$ | $59.39_{\pm 8.7}$ | $66.47_{\pm 9.9}$ | $60.83_{\pm 8.6}$ |
| GIGAP [64] (1134M Para.) (Slide-170K) | ABMIL | $96.43_{\pm 3.5}$ | $71.85_{\pm 2.1}$ | $94.39_{\pm 3.4}$ | $96.54_{\pm 1.7}$ | $60.56_{\pm 8.6}$ | $63.81_{\pm 9.3}$ | $59.85_{\pm 8.1}$ |
| | TransMIL | $96.59_{\pm 3.2}$ | $65.45_{\pm 2.0}$ | $93.97_{\pm 3.9}$ | $97.61_{\pm 1.2}$ | $60.40_{\pm 8.8}$ | $62.90_{\pm 9.2}$ | $60.12_{\pm 8.5}$ |
| | GIGAP | $95.53_{\pm 3.1}$ | $65.86_{\pm 2.2}$ | $93.72_{\pm 3.4}$ | $97.53_{\pm 1.2}$ | $62.99_{\pm 8.7}$ | $62.64_{\pm 9.3}$ | $57.63_{\pm 5.4}$ |
| | ABMILX | $96.74_{\pm 3.0}$ | $73.01_{\pm 2.1}$ | $94.83_{\pm 3.5}$ | $96.09_{\pm 2.0}$ | $59.69_{\pm 8.7}$ | $66.34_{\pm 8.8}$ | $57.81_{\pm 8.4}$ |
| *E2E Approaches* | | | | | | | | |
| ResNet-18 | ABMILX | $95.88_{\pm 2.7}$ | $78.34_{\pm 0.6}$ | $93.97_{\pm 2.9}$ | $97.09_{\pm 1.4}$ | $64.91_{\pm 8.7}$ | $67.78_{\pm 8.8}$ | $61.20_{\pm 8.0}$ |
| ResNet-50 | ABMILX | $96.06_{\pm 2.3}$ | $78.83_{\pm 0.6}$ | $95.17_{\pm 2.8}$ | $97.06_{\pm 1.5}$ | $64.72_{\pm 8.4}$ | $67.20_{\pm 8.6}$ | $60.78_{\pm 8.4}$ |

# C  Additional Quantitative Results

## C.1  More about E2E Methods

Slide-level supervised E2E methods process entire slides in a unified manner, preserving the critical interdependencies necessary for robust clinical interpretation. These methods exploit the complete spatial context of gigapixel images, thereby providing a more comprehensive and clinically pertinent analysis. Building on these merits, several researchers have introduced innovative slide-level supervised E2E methods to further enhance WSI analysis. Sharma et al. [50] proposed an E2E framework (C2C) that clusters patch representations and employs adaptive attention with KL-divergence regularization to robustly classify whole slide images. Li et al. [30] proposed a task-specific fine-tuning framework (FT) that employs a variational information bottleneck to distill patches into a sparse subset via Monte Carlo-sampled Bernoulli masks, thereby enabling E2E backbone fine-tuning.

In this section, we compare and analyze these E2E methods. As shown in Table 6, existing E2E pipelines often allocate substantial computational resources to patch or region-level sampling strategies for processing gigapixel WSIs, resulting in prolonged training times (e.g., 84h for C2C [49] and 45h for FT [61]), yet yielding only marginal performance gains compared to two-stage approaches with R50 features. Once FMs are integrated into two-stage frameworks, the FM-based approach outperforms previous E2E methods by a significant margin, effectively diminishing their advantage. In contrast, our ABMILX approach drastically shortens training time (9h with ResNet18, which is comparable to SOTA two-stage ResNet50) and simultaneously delivers substantial performance improvements over both prior E2E and two-stage ResNet50-based methods. Moreover, even when compared with latest two-stage methods under FM features, ABMILX-R50 maintains competitive performance in both training efficiency and final performance, highlighting the effectiveness of our approach.

| | Grad. | Sub. | Surv. |
|---|---|---|---|
| w/o FFN | 77.04 | 93.97 | 67.78 |
| w/ FFN | 78.34 | 93.03 | 65.23 |

(a) **Feed Forward Network.**

| | Grad. | Sub. | Surv. |
|---|---|---|---|
| 128 | 76.22 | 91.50 | 62.64 |
| 256 | 76.15 | 93.97 | 67.78 |
| 384 | 76.01 | 90.21 | 64.13 |
| 512 | 78.34 | 91.51 | 63.20 |

(b) **Projection Dim** for input of ABMILX.

| | Grad. | Sub. | Surv. |
|---|---|---|---|
| w/o MH. | 73.74 | 90.12 | 63.57 |
| 2 | 77.32 | 90.91 | 65.61 |
| 4 | 78.34 | 91.65 | 62.83 |
| 8 | 76.48 | 93.97 | 67.78 |
| 16 | 76.50 | 92.59 | 64.62 |

(c) **Head Number** in Multi-head Local Attention of ABMILX. MH. denotes multi-heads.

| | Grad. | Sub. | Surv. |
|---|---|---|---|
| w/o MS. | 76.77 | 92.72 | 67.24 |
| 2 | 74.02 | 92.74 | 64.92 |
| 4 | 78.34 | 93.97 | 66.22 |
| 6 | 77.21 | 92.35 | 67.46 |
| 10 | 76.09 | 92.25 | 67.78 |

(d) **Multi-scale Ratio** in Multi-scale Random Sampling. MS. denotes multi-scale.

| | Grad. | Sub. | Surv. |
|---|---|---|---|
| 64 | 76.22 | 93.07 | 63.52 |
| 128 | 78.34 | 93.86 | 63.73 |
| 384 | 74.32 | 92.52 | 64.12 |
| 512 | 75.62 | 93.97 | 65.59 |
| 768 | - | 93.66 | 67.78 |
| 1280 | - | 92.23 | 65.77 |

(e) **Sampling Number** in Multi-scale Random Sampling.

| | Grad. | Sub. | Surv. |
|---|---|---|---|
| Rand. | 78.34 | 93.97 | 67.78 |
| *Regional Rand. Sampling* | | | |
| 2 | 76.52 | 93.26 | 67.28 |
| 4 | 76.11 | 92.67 | 67.23 |
| 8 | - | 92.14 | 67.00 |

(f) **Sampling Strategy**. Rand. denotes naive random sampling.

Table 9: Ablation studies on various components of our method. Default settings are marked in gray .

## C.2 More about MILs in E2E Learning

Table 7 presents the performance of various MIL aggregators in the E2E learning. Besides the commonly used DSMIL [29] and TransMIL [48], we implemented ViTMIL, based on a two-layer multi-head self-attention (MSA) structure, and QAMIL, using a single-layer multi-head query attention. Comparing existing sparse attention methods and Transformer-like methods, we observe: 1) The E2E optimization challenge posed by sparse attention is significant. Specifically, although RRTMIL [56], TransMIL, and ViTMIL share a similar MSA front-end structure, the difference in their final aggregation methods leads to substantial performance variations. RRTMIL directly employs ABMIL as the aggregator, while the other two utilize the [CLS] token from the MSA. This demonstrates the impact of sparse attention on encoder optimization. Furthermore, the feature re-embedding module (MSA layer) in RRTMIL further impairs the representation of the affected features, accelerating the collapse of the optimization loop. 2) Maintaining sparsity is beneficial for E2E optimization. Although we experimented with various Transformer-like methods in the E2E setting, they still underperformed compared to FM. We attribute this to the fact that in the E2E training, relying solely on global attention struggles to focus on learning key regions within the numerous redundant patches. Our proposed ABMILX mitigates the E2E optimization challenges while preserving sparsity, thus achieving superior performance.

## C.3 ABMILX in Two-Stage Framework

We supplement quantitative results on the CAMELYON [4, 3] dataset in Table 8 and demonstrate ABMILX's performance in a two-stage framework. The results reveal the following insights: 1) The quality of offline features determines the performance of two-stage methods. Although different MIL aggregators show performance variations, these differences are significantly smaller under FMs compared to ResNet-50 (R50). Under FM features, classical ABMIL often outperforms advanced MIL approaches, highlighting the importance of sparse attention in CPath tasks. 2) Two-stage methods based on FM features have achieved saturated performance on traditional tasks (CAMELYON and NSCLC), leveraging massive pre-training data and large models. Further performance gains from increased data volume and model size are limited. Conversely, large FMs also encounter performance bottlenecks in challenging tasks (BRCA-subtyping and Survival Analysis). We attribute this to the encoder's lack of downstream task adaptation. Our proposed E2E method surpasses performance on these challenging benchmarks, demonstrating the effectiveness and potential of E2E approaches in CPath tasks. 3) We further validate the effectiveness of ABMILX in two-stage paradigms, demonstrating its versatility as a pure MIL architecture in CPath tasks. ABMILX performs

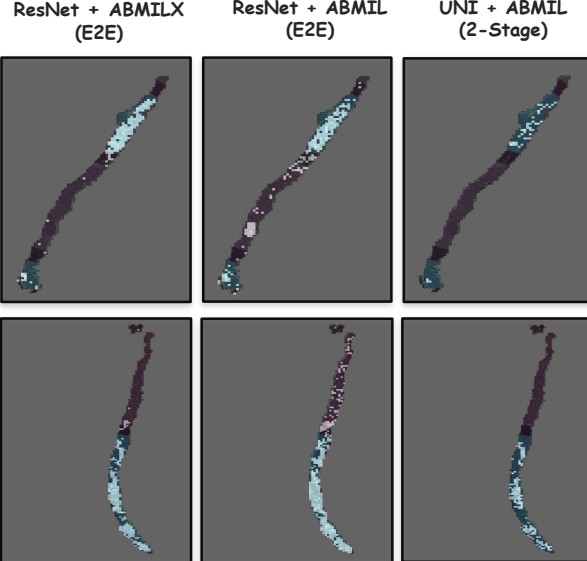

|ResNet + ABMILX (E2E) | ResNet + ABMIL (E2E) | UNI + ABMIL (2-Stage)|

Figure 4: Attention visualization on the PANDA dataset [6]. All slides are from the Karolinska Center, with annotations limited to three types: background, benign tissue, and cancerous tissue. We highlight cancerous tissue in blue and display high-attention patches as bright patches for comparison.

more exceptionally under FM features, further confirming the significance of maintaining sparse characteristics for CPath tasks.

### C.4 Ablation

We conduct ablation studies on the hyperparameters related to our method in Table 9 and provide the following analysis.

**Feed Forward Network.** Feed Forward Network (FFN) is a common component in modern models [35, 36, 18], which we explore in ABMILX's design. We add FFN after the bag feature aggregation module to further refine bag features before input to the task head. We find that due to FFN's large parameter count, it requires a larger training dataset. It performs poorly on conventional datasets with smaller data scales, except for PANDA. PANDA, with its 10,000 slides, allows FFN to demonstrate performance improvements.

**Sampling Number.** Sampling number is a critical hyperparameter in sampling strategies, closely related to downstream tasks. We observe that a larger sampling number does not consistently improve performance. We attribute this to the fact that an appropriate sampling number helps the model eliminate redundant instances that interfere with optimization. For PANDA, with an average of around 500 patches, lower sampling numbers perform better. In contrast, survival analysis tasks often require a larger sampling number for more comprehensive analysis.

**Sampling Strategy.** Beyond naive random sampling, we explore region-based random sampling. We divide the entire slide into N sub-regions and perform random sampling within each sub-region. The results in Table 9f demonstrate that the diversity brought by naive random sampling is beneficial for optimization. However, more region division can compromise diversity and lead to performance degradation.

## D  Additional Qualitative Results

**PANDA.** To further demonstrate the effectiveness of ABMILX and E2E learning, we visualize the attention scores (bright patches) of different methods in Figure 4. We demonstrate the following: (1) Compared to the offline FM approach, the encoder trained with E2E (column 1 and 2) produces more cohesive instance features. This enables the MIL attention to more comprehensively cover cancerous tissue (blue areas). (2) Compared to ABMIL, ABMILX reduces attention to normal patches. This

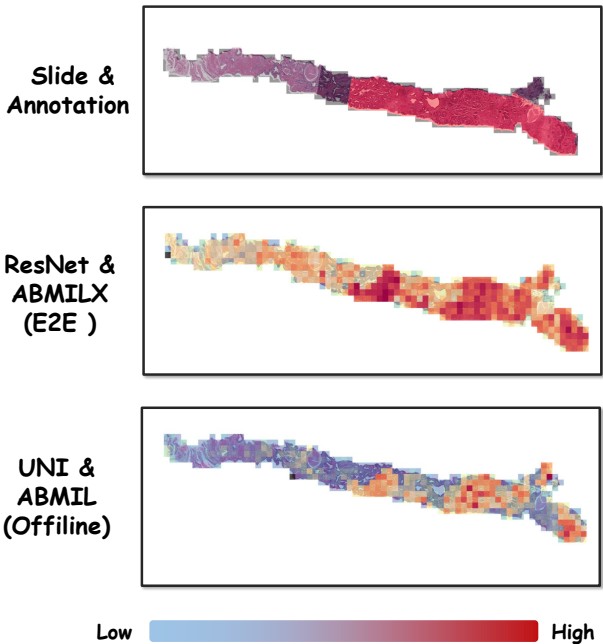

Figure 5: Heatmap visualization on the PANDA dataset [6]. The top row shows original slide and its annotation (with cancerous tissue in red). The middle and bottom rows present attention maps generated by *ResNet & ABMILX (E2E)* and *UNI & ABMIL (Offline)* respectively. Color intensity ranges from blue (low attention) to red (high attention), illustrating how each approach prioritizes different tissue regions. Notably, our model yields a more uniform attention distribution while effectively highlighting cancerous areas.

indicates that ABMILX allows the encoder to learn more effectively from discriminative patches in the E2E training. It leads to more accurate representations of normal patches and improved differentiation from tumor patches. (3) Beyond improved feature, ABMILX also benefits from the proposed global attention plus module. This module refines the raw attention map based on feature correlations, mitigating the issues of excessive attention to redundant regions and insufficient attention to discriminative regions.

**CAMELYON.** We provide additional visualizations of different MIL models during E2E training and convergence stage in right Figure. During E2E training, extreme sparsity causes ABMIL [25] to overlook discriminative regions while overly focusing on redundant areas. Although TransMIL [48] covers a small number of discriminative regions, it is distracted by a large amount of attention on redundant ones. This prevents the encoder from adequately learning discriminative regions, causing ABMIL to fail in correctly localizing target areas. While the converged TransMIL can localize it, its training process struggles to consistently focus on discriminative areas, resulting in incomplete identification of the overall tumor region. In contrast, ABMILX benefits from more effectively enabling the encoder to learn from discrimina-

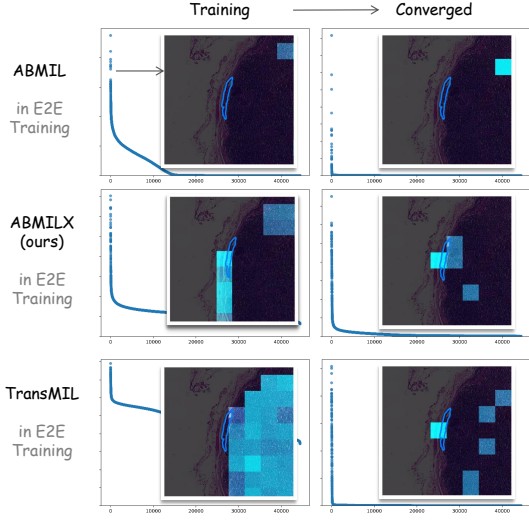

tive regions during training. Consequently, it achieves stronger discriminative capabilities in converged stage, simultaneously enhancing focus on tumor regions while reducing interference from redundant areas.

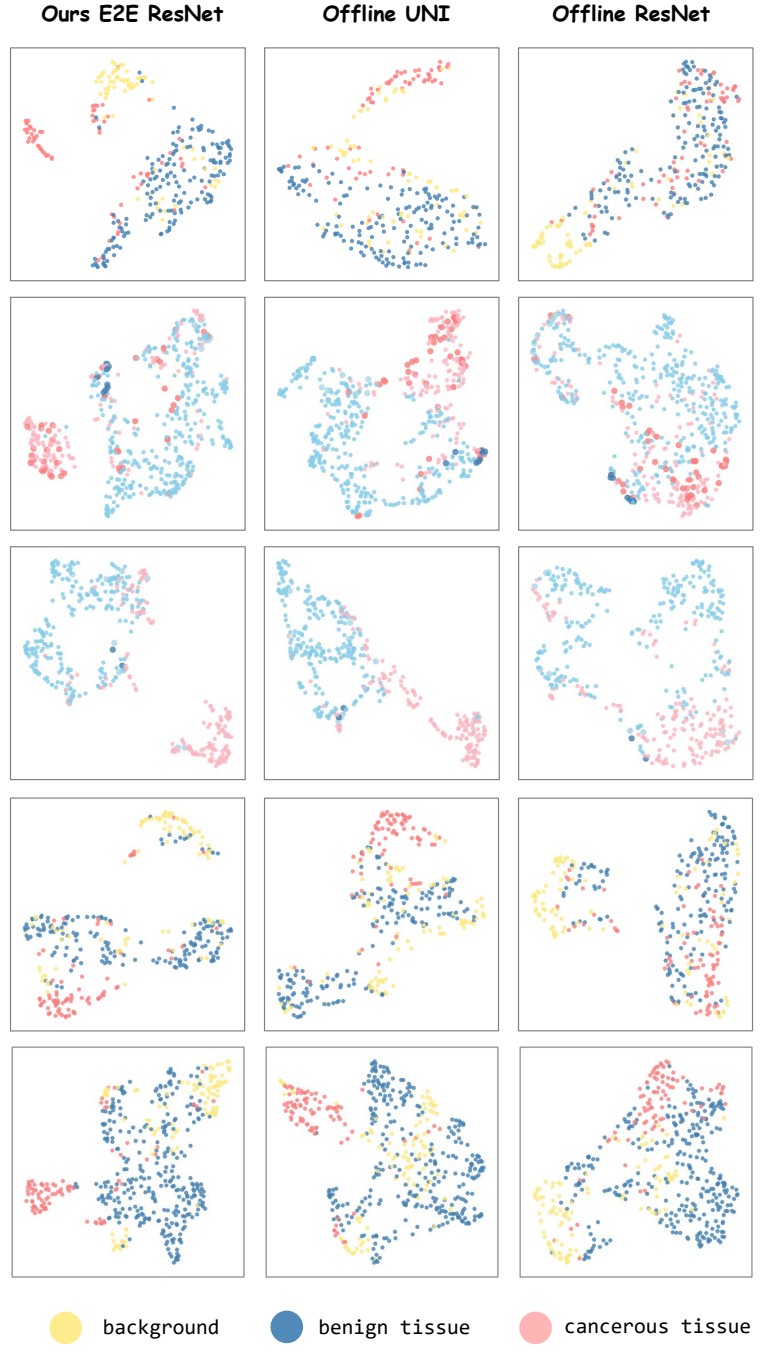

Figure 6: More UMAP [22] visualization on PANDA dataset [6].

**More Visualization.** To provide a more comprehensive quantitative analysis of the proposed method, we present heatmaps and additional UMAP [22] visualizations in Figures 5 and 6, respectively.

# E Additional Related Works

While E2E high-resolution image analysis is relatively mature in general computer vision [45, 44, 5], its application to CPath presents significant challenges due to their unique gigapixel scale. As elaborated in the *Related Work* section, current E2E methods for WSI analysis are broadly categorized into instance-level supervised and slide-level supervised approaches. As elaborated in the *Related*

*Work* section, current E2E methods can be divided into instance-level supervised approaches and slide-level supervised approaches. Instance-level supervised methods [13, 39, 46, 47] adopt a pseudo E2E paradigm, in which the encoder is trained using instance-level pseudo-labels rather than genuine slide-level supervision. This strategy simplifies the problem by processing patches independently; however, it neglects the essential inter-patch contextual relationships required for robust clinical interpretation [7, 40] and creates a disconnect between training and downstream clinical applications. Moreover, the performance of these methods fundamentally depends on the quality of the pseudo-labels, indicating that they constitute a compromise rather than a fully E2E solution. For example, Qu et al. [47] introduce a novel instance-level MIL framework that leverages weakly supervised contrastive learning and prototype-based pseudo label generation to markedly improve both instance and bag-level classification in WSI analysis. Luo et al. [39] propose a negative instance guided self-distillation framework that leverages true negative samples and a prediction bank to constrain pseudo-label distributions, enabling an E2E instance-level classifier.

## F   Limitation & Broader Impacts

This work pioneered the exploration of end-to-end (E2E) optimization challenges in computational pathology and effectively mitigated them. It demonstrated the potential and advantages of E2E learning in this domain. However, our current full-training approach makes direct fine-tuning of large foundation models challenging under limited computational resources. Investigating the effectiveness of our proposed method for fine-tuning foundation models is a direction for future work. Furthermore, as this work focuses on computational pathology, it is directly relevant to tasks such as multi-cancer diagnosis and prognosis. This work has the potential to inspire and facilitate the deployment of more accurate and efficient clinical diagnosis and prognosis algorithms.

