# OpenReview forum: "Revisiting End-to-End Learning with Slide-level Supervision in Computational Pathology"
_NeurIPS.cc/2025/Conference — NeurIPS 2025 poster_

### Official Review · Reviewer_igAe · 2025-06-30

**Clarity:** 3
**Significance:** 3
**Originality:** 3
**Rating:** 4
**Confidence:** 4

**Summary:**

This paper revisits slide-level supervised end-to-end (E2E) learning in computational pathology,
and proposes a novel MIL model, ABMILX, to address the optimization challenge caused by sparse-attention MIL.
The proposed ABMILX integrates multi-head attention and global correlation-based refinement,
combined with multi-scale random sampling.
The framework proposed in the paper achieves superior performance across multiple benchmark datasets.

**Questions:**

1. From Table 1,
the performance gap between UNI + ABMIL and E2E + ResNet50 + ABMILX does not appear to be very large.
However, according to Fig. 1(a), E2E + ResNet shows a clear improvement over models like UNI.
Does this suggest that the performance difference between ABMILX and ABMIL is actually small?
I believe the authors need to provide more comparisons under fair conditions to further evaluate the method.

2. In addition to being used as a standalone prediction model, ABMIL has also been widely applied in other frameworks,
for example, as a pooling tool after GNN message passing (e.g. PatchGCN[1]).
How does ABMILX perform when replacing ABMIL in these settings?

3. In Table 3, regarding the results of models under the two-stage paradigm, were patches also randomly sampled? I would like to see further comparisons of ABMILX with more recent state-of-the-art models under two-stage settings.


Although I do not fully agree with the paper’s position on E2E, I really like the design of ABMILX. I think ABMILX is an interesting and easily generalizable model, and I am therefore particularly interested in seeing more comprehensive experiments on ABMILX.
I am willing to increase my ratings based on the rebuttal.


[1] Chen, Richard J., et al. "Whole slide images are 2d point clouds: Context-aware survival prediction using patch-based graph convolutional networks", MICCAI 2021

**Ethical Concerns:**

["NO or VERY MINOR ethics concerns only"]

**Final Justification:**

While I still have concerns regarding the scalability of the proposed end-to-end approach, I acknowledge the contribution of ABMILX. The additional experiments presented in the rebuttal suggest that ABMILX is promising.

**Limitations:**

yes

**Quality:**

2

**Strengths And Weaknesses:**

**Strengths**

This paper proposes ABMILX to mitigate the optimization risks from both local and global perspectives.
I find this an interesting and easily generalizable module that could inspire future work.
ABMILX computes patch attention scores using self-attention,
allowing the model to assign patch importance with global awareness.
It also leverages multiple replicas and multi-head attention to capture diverse features.
I think this design very intuitive, and I have not seen a similar approach before.

The overall idea is easy to follow, and the method is clearly explored.
The figures and discussions are well done.

**Weaknesses**

I have some concerns regarding the E2E part.

I do not fully agree with the paper's description of E2E.
Pathology foundation models have already been widely developed.
Although pretraining these models on large-scale datasets comes at a high cost,
but once trained, they can be applied directly to most datasets and tasks without fine-tuning,
and have been proven to perform well across various cancer types and tasks.
While E2E encoders might be able to extract more task-specialized features,
the need to train separate models for different datasets and tasks could actually increase overall training cost.

In addition, I feel that implementing E2E by using a smaller encoder (only ResNet) and randomly sampling a small number of patches is rather crude. Since E2E requires random patch sampling, it becomes difficult to properly model relationships between patches (such as spatial positioning or multi-scale structures), which many studies have shown to be crucial. This also limits the scalability of the E2E framework, making it challenging to design models that can capture richer contextual information.

Although I really like the design of ABMILX, I think the exploration of ABMILX in this paper is insufficient. The comparative methods chosen are relatively outdated. I would like to see combinations of advanced backbone encoders with recent aggregation strategies, such as WIKG + UNI. It is unclear why these newer models were only tested in a two-stage setting with a ResNet50 encoder. I hope to see more comparisons with state-of-the-art models from the past two years, especially under settings that use strong foundation model encoders.

---

> ### Author Rebuttal · Authors · 2025-07-31
>
> Thanks for providing feedback and taking the time to review our work! We have provided point-by-point clarifications on the issues and supplemented the experiments as suggested.
>
> > **W1&W2: Clarification on E2E and Foundation Model**
>
> **R1:**
>
> - First, we would like to clarify that E2E and Pathology Foundation Models (FM) are not opposing concepts. We acknowledge that FMs have made extensive applications and prominent contributions in pathology due to their effectiveness and generalization. However, we argue that the performance of traditional patch-based FMs is approaching saturation [1], and an increasing number of FMs are adopting E2E-based self-supervised pre-training [2, 3]. **We believe that E2E with slide-level supervision primarily advantages in its training process being more aligned with clinical tasks** (rather than merely learning from medical images themselves) and **its more efficient data utilization**. A concurrent work holds a similar view [4]. We have shown that small ResNet models pre-trained only on ImageNet-1k can achieve performance comparable to or even superior to that of two-stage schemes based on computationally expensive pre-trained FMs, with **E2E training costing <10 3090 GPU-hours (a cost similar to, or even lower than, the training cost of some latest large-scale slide encoders** [3]).
>
> - More importantly, we supplement the following **external validation** to demonstrate the generalization ability of encoders trained via E2E:
>
>   - | TCGA->CPTAC          | R50+ABMIL | R50+TransMIL | R50+WIKG | UNI+ABMIL | UNI+TransMIL | UNI+WIKG | R50+ABMILX (E2E) |
>     | -------------------- | --------- | ------------ | -------- | --------- | ------------ | -------- | ---------------- |
>     | CPTAC-NSCLC (AUC)    | 66.42     | 74.59        | 64.04    | 83.73     | 85.24        | 83.56    | 85.19            |
>     | CPTAC-LUAD (C-index) | 46.34     | 48.24        | OOM      | 53.59     | 51.36        | OOM      | 54.00            |
>
> *OOM = Out-of-Memory on 24GB-RTX-3090*
>
> Without any fine-tuning, the model trained on TCGA was directly tested on CPTAC-sourced data. **The R50 trained via E2E not only showed significantly improved generalization but even outperformed UNI.**
>
> - Furthermore, constrained by resources, this work is limited to small-scale models and datasets. However, we believe that E2E, after addressing optimization challenges, has greater potential in pathology. **We conducted a preliminary attempt at UNI by applying ABMILX for E2E Parameter-Efficient Fine-Tuning (PEFT) on UNI.** The following results demonstrate the advantages of ABMILX in E2E tasks and preliminarily verify the potential of E2E to further enhance FMs. We note that work [5] adopted a similar approach but overlooked the impact of optimization challenges caused by MIL on E2E training.
>
>   - |       | UNI   | R50 (E2E) | UNI+PEFT (E2E) | UNI+PEFT (E2E) | UNI+PEFT (E2E) |
>     | ----- | ----- | --------- | -------------- | -------------- | -------------- |
>     |       | ABMIL | ABMILX    | ABMIL          | TransMIL       | ABMILX         |
>     | PANDA | 74.69 | 78.83     | 74.11          | 76.61          | 80.99          |
>
> - 【**Patch Sampling**】 In fact, we **only perform patch sampling during the training phase, with sampling varying across iterations to ensure diversity.** This approach is relatively common not only in general domains [6] but also in Pathology [7]. We have conducted comprehensive ablation studies on the sampling strategy in both the main text and supplementary materials, and found that it not only achieves optimal efficiency but also helps eliminate data redundancy and enhance data diversity.
>
> > **W3&Q3: Clarification on Latest Baselines and Two-stage Details**
>
> **R2:** In fact, **we compared the performance of WIKG and RRT under FM features (UNI, CHIEF, GIGAP) in Table 1 of the Supplemental Material.** In the table below, we have reorganized these results and **supplemented 2DMamba [8]**. In the revised version, this content will be included in the main text. Furthermore, except for the training of the GIGAP aggregator (due to memory constraints), all other two-stage experiments did not employ any random sampling strategy. Additional details are provided in Supplemental Material B.3.
>
> | Encoder   | Method  | Grading   | Subtyping | Survival  |
> | --------- | ------- | --------- | --------- | --------- |
> | R18 (E2E) | ABMILX  | 78.34     | 93.97     | **67.78** |
> | R50 (E2E) | ABMILX  | **78.83** | **95.17** | 67.20     |
> | UNI       | RRT-MIL | 74.00     | 94.61     | 66.91     |
> | UNI       | WIKG    | 74.97     | 94.19     | 65.28     |
> | UNI       | 2DMamba | 74.97     | 93.59     | OOM       |
> | CHIEF     | RRT-MIL | 69.73     | 92.49     | 67.30     |
> | CHIEF     | WIKG    | 71.22     | 93.13     | 65.92     |
> | CHIEF     | 2DMamba | 71.59     | 90.96     | OOM       |
> | GIGAP     | RRT-MIL | 72.46     | 94.84     | 65.40     |
> | GIGAP     | WIKG    | 72.17     | 94.76     | 66.97     |
> | GIGAP     | 2DMamba | 75.36     | 93.33     | OOM       |
>
> *OOM = Out-of-Memory on 24GB-RTX-3090*
>
> > **Q1&Q2: ABMIL vs. ABMILX on more applications**
>
> **R3:** **Multiple fair ablation studies comparing ABMILX and ABMIL are presented in both the main text (Figure 1(c), Sec.3.3, Sec.4.3) and supplemental material (Table 2, Table 3, Sec.D)**. Here, we reorganize and summarize their comparisons in E2E and Two-Stage settings, and **supplement the comparisons in more scenarios** (Pavement Distress Classification, Diabetic Retinopathy Grading, Path GCN framework).
>
> - **【Clarification of details】** The data in Figure 1(a) are identical to those in Table 1. *E2E+ResNet* in Figure 1(a) corresponds to *E2E+ResNet-50+ABMILX* in Table 1. The possible reason for the minor discrepancy is that Table 1 only includes one challenging benchmark, while Figure 1 is more comprehensive.
>
> - **【E2E & Two-Stage】** We have summarized the table below from the main text and supplemental material. All comparisons in the table ensure that all other settings are identical except for the MIL model.
>
>   - |        | E2E     | E2E       | E2E      | Two-Stage | Two-Stage | Two-Stage |
>     | ------ | ------- | --------- | -------- | --------- | --------- | --------- |
>     |        | Grading | Subtyping | Survival | Grading   | Subtyping | Survival  |
>     | ABMIL  | 75.46   | 89.23     | 62.70    | 71.85     | 94.39     | 63.81     |
>     | ABMILX | 78.34   | 93.97     | 67.78    | 73.01     | 94.83     | 66.34     |
>
> - **【More Tasks】** Following [9], we supplemented experiments on E2E recognition of high-resolution images, such as Pavement Distress Classification (CQU-BPDD [10]) and Diabetic Retinopathy Grading [11], to more thoroughly compare ABMIL and ABMILX.
>
>   - | Task\Model | ABMIL | TransMIL | ABMILX |
>     | ---------- | ----- | -------- | ------ |
>     | CQU-BPDD   | 81.3  | 82.0     | 82.5   |
>     | DRG        | 75.3  | 74.3     | 78.1   |
>
> - **【PathGCN】** We supplemented comparisons with PathGCN, demonstrating the effectiveness of ABMILX when applied to other frameworks.
>
>   - | PathGCN+\Task | Grading | Subtyping | Survival |
>     | ------------- | ------- | --------- | -------- |
>     | ABMIL         | 76.90   | 93.64     | 64.97    |
>     | ABMILX        | 77.92   | 94.23     | 65.96    |
>
> [1] PathBench: A comprehensive comparison benchmark for pathology foundation models towards precision oncology.
>
> [2] A Multimodal Knowledge-enhanced Whole-slide Pathology Foundation Model.
>
> [3] A whole-slide foundation model for digital pathology from real-world data. Nature.
>
> [4] Foundation Models - A Panacea for Artificial Intelligence in Pathology?
>
> [5] Unlocking adaptive digital pathology through dynamic feature learning.
>
> [6] Masked Autoencoders Are Scalable Vision Learners. CVPR 2022.
>
> [7] Multiple instance learning framework with masked hard instance mining for whole slide image classification. ICCV 2023.
>
> [8] 2DMamba: Efficient State Space Model for Image Representation with Applications on Giga-Pixel Whole Slide Image Classification. CVPR 2025.
>
> [9] No Pains, More Gains: Recycling Sub-Salient Patches for Efficient High-Resolution Image Recognition. CVPR 2025.
>
> [10] An iteratively optimized patch label inference network for automatic pavement distress detection. T-ITS.
>
> [11] Diagnostic assessment of deep learning algorithms for diabetic retinopathy screening. Information Sciences.

---

> > ### Comment · Reviewer_igAe · 2025-08-02
> >
> > Thank you for your responses and clarifications.
> >
> > While I agree that patch-based FMs is approaching saturation,
> > my primary concern lies in the scalability of the proposed E2E approach based on random sampling and a lightweight encoder.
> > As noted in my review, the scalability of this design remains questionable.
> > The cited works \[2]\[3] adopt frozen patch encoders for slide-level tasks and no patch sampling,
> > which is not aligned with the E2E formulation claimed here.
> >
> > Regarding patch sampling, I would like to see how an E2E-trained encoder performs with more complex aggregation methods (e.g., PatchGCN) v.s. FM encoders,
> > to better validate E2E's generalizability.
> >
> > I appreciate the authors’ efforts in rebuttal.
> > Overall, I consider the paper to be at a marginal level, and I am inclined to accept it at the marginal level.
> > I have updated my score to 4,
> > and encourage the authors to further strengthen the work along the above lines.

---

> ### Author Response · Authors · 2025-08-03
> **Response by Authors**
>
> Thank you for your appreciation and suggestions.
> However, we would still like to make some clarifications regarding the framework issue and the scalability of E2E learning.
> First of all, what we would like to clarify is that our method is not limited to random sampling or small-scale encoders. Constrained by resources, we adopted a combination of small-scale encoders and random sampling. However, its actual performance is surprising, which is highly consistent with the viewpoint of [4]. This work also achieved performance close to that of foundation model (FM) encoders by end-to-end (E2E) training small-scale encoders with random sampling, challenging the current high-cost foundation model pretraining. Regarding works [2, 3] cited in our rebuttal, we would like to clarify that although both serve as frozen feature encoders in downstream tasks, [2] indeed utilized random sampling and slide aggregators in its pretraining phase to optimize the patch encoder.  We suggest this can be regarded as an application of E2E in pretraining.
> As for sampling strategies, we have actually discussed them sufficiently in the manuscript. We consider the choice of sampling strategy as a trade-off between performance and efficiency. In the reorganized table below, we investigated the impact of different sampling strategies on E2E training and found that while different sampling strategies have little impact on performance, they significantly affect training efficiency. Moreover, random sampling does not actually reduce performance significantly. Our proposed MRIS, after supplementing multi-scale information to alleviate the problem of the model's local vision that may be caused by random sampling, achieves even better performance.
>
> Task/Method          | ABMILX + Attention | ABMILX + MRIS | ABMILX + RS | ABMIL + RS | ABMIL + MRIS | ABMIL + Attention
> ---------------------|--------------------|---------------|-------------|------------|--------------|-------------------
> Survival             | 66.53              | 67.78         | 67.24       | 62.64      | 62.70        | 62.74
> Subtyping            | 93.14              | 93.97         | 92.72       | 87.99      | 89.23        | -
> Grading              | 77.43              | 78.34         | 76.77       | 73.51      | 75.46        | -
> Train Time           | 68h                | 9h            | 9h          | 6h         | 6h           | -
>
> Furthermore, if further reduction of E2E memory consumption is required (e.g., when training foundation models), we have also validated the use of recently proposed dual-buffer sampling—wherein a subset of patches from those used in forward propagation is selected for backward propagation. ABMILX and the E2E framework can still achieve substantial performance while maintaining extremely low memory consumption.
>
> Task/Method               | R50 + ABMILX + MRIS | R50 + ABMILX + Dual-Buffer
> --------------------------|---------------------|------------------------------
> Grading                   | 78.38               | 76.34
> Subtyping                 | 95.17               | 92.71
> Memory                    | 24g                 | 6.7g
>
> On the selection of different encoders: to demonstrate the scalability of our framework and method, we have replaced the small-scale encoder with the foundation model UNI in this rebuttal. This is a preliminary attempt, where we applied MRIS and ABMILX for PEFT on UNI. Results in the table below preliminarily indicate that the advantages of MRIS and ABMILX on small-scale models can be extended to foundation models. ABMILX still achieves significant performance gains by mitigating optimization challenges, while MRIS also exhibits stable performance (The computational cost would be prohibitive if more complex sampling strategies were adopted in end-to-end training with foundation-level encoders, whereas MRIS only required ~40 3090 GPU hours).
>
> Encoder           | MIL      | PANDA
> ------------------|----------|------
> UNI               | ABMIL    | 74.69
> R50 (E2E)         | ABMILX   | 78.83
> UNI+PEFT (E2E)    | ABMIL    | 74.11
> UNI+PEFT (E2E)    | TransMIL | 76.61
> UNI+PEFT (E2E)    | ABMILX   | 80.99
>
> Additionally, we are following the reviewers' suggestions to explore the performance of E2E trained encoders compared to FM under MIL methods such as PathGCN. However, we would like to emphasize that the experiments in our supplementary material—where ABMILX is applied to two-stage FM features—demonstrate that its performance is highly competitive. We consider it to be an advanced MIL method.

---

> ### Author Response · Authors · 2025-08-03
> **Response by Authors**
>
> When the performance of the two-stage paradigm is approaching its bottleneck,  E2E in computational pathology remains a vast uncharted territory. Its true prosperity, popularity, and generalization will inevitably require the participation of researchers across the entire community in the future. As stated in our title, this work aims to revisit E2E while exploring its potential and challenges to the best of our ability, and we suggest we have made every effort to achieve this. We would like to once again express our gratitude for your patience, appreciation, and suggestions.

---

> ### Author Response · Authors · 2025-08-09
>
> Thank you again for the helpful discussion. As the deadline is approaching, we just wanted to quickly and respectfully check if our most recent response has resolved your remaining concerns. Please let us know if anything remains unclear.

---

> > ### Comment · Reviewer_igAe · 2025-08-09
> >
> > Thank you for the detailed feedback, which I have carefully reviewed.
> > Overall, the authors have effectively addressed most of the issues raised.
> >
> > I agree that E2E is a field with great potential for development.
> > However, although the authors claim that their method is not limited to random sampling or small-scale encoders,
> > I believe it is necessary to further discuss different forms of E2E in the paper.
> > For instance, \[2] essentially uses a frozen encoder to train the aggregator and additionally uses a frozen aggregator to train the encoder,
> > which can indeed be regarded as a generalized form of E2E (optimizing patch encoder according to the task),
> > but differs from the E2E proposed in this paper.
> > (the narrow-sense E2E is generally refers to the simultaneous end-to-end training of both the aggregator and encoder, against two-stage, and the scalability of this mode may be limited)
> >
> > As mentioned earlier, I am particularly concerned about the scalability of the E2E framework (including its ability to extend to different aggregators and encoders).
> > It is also recommended that the authors conduct a more in-depth discussion on this in the final version, which will significantly enhance the completeness and value of the research.
> >
> > In summary, I believe this paper meets the acceptance criteria. Thank you for the careful clarifications.

---

> > > ### Author Response · Authors · 2025-08-09
> > > **Acknowledgements by Authors**
> > >
> > > We understand your penetrating insights and greatly value your excellent suggestions. Building on this work, which has explored the impact of the MIL architecture on the effectiveness of E2E, our future research will continue to investigate the impact of the E2E framework design on corresponding generalization.

---

### Official Review · Reviewer_tziM · 2025-07-03

**Clarity:** 3
**Significance:** 2
**Originality:** 2
**Rating:** 4
**Confidence:** 4

**Summary:**

The paper argues that the traditional two-stage learning paradigm lacks joint optimization. It suggests that this process is limited by an optimization collapse, which is caused by the interaction between the encoder and the MIL module. To solve this optimization collapse problem, a new MIL aggregator is proposed, composed of multi-head local attention and global correlation-based attention. At the same time, to improve the effectiveness of the implementation, a multi-scale random sampling strategy is also applied in the model.

**Questions:**

1. What are the separate contributions of the ABMILX module and the multi-scale sampling strategy to the performance
2. The paper's baselines are somewhat old. Please find more recent and novel baselines for comparison.
3. The authors are requested to provide a more rigorous mathematical analysis explaining how sparse attention causes optimization collapse in the E2E setup. Additionally, a novel metric to quantify optimization collapse, along with a visual representation of the separate contributions of the sampling strategy and ABMILX in addressing this collapse, would be beneficial.

**Ethical Concerns:**

["NO or VERY MINOR ethics concerns only"]

**Final Justification:**

Thank you for the author's detailed explanation. All my questions about the innovation and experimental part have been answered.

**Limitations:**

see Questions and Weaknesses

**Quality:**

2

**Strengths And Weaknesses:**

Strengths:
1.The primary contribution is not just the proposed model (ABMILX), but the pioneering identification and clear elucidation of the optimization challenge caused by sparse-attention MIL in an E2E setting. Figure 2 provides an excellent conceptual illustration of this "deteriorating iteration" problem.
2.A MIL aggregator is proposed, based on multi-head local attention and global correlation-based attention , which improves the model's performance by attempting to solve the problem of optimization collapse in joint optimization.
Weaknesses:
1. This paper's main contribution is proposing a new MIL aggregator that utilizes different attention mechanisms; however, the innovation is very limited and the contribution is insufficient. Additionally, the design of the attention mechanisms used is very common and lacks theoretical support.
2. The authors use a combination of a sampling strategy and ABMILX to improve the model's performance. The ablation study does not specify the respective contributions of the sampling strategy and the ABMILX module to the model. Meanwhile, while the ablation study compares two different sampling strategies, the comparison is not convincing.

---

> ### Author Rebuttal · Authors · 2025-07-31
>
> > **W1: Novelty and contribution**
>
> **R1:**
>
> 【**Innovation in research direction**】We would like to clarify that the main contribution of this paper lies in **Revisiting slide-level E2E work in computational pathology** and proposing that the **overlooked sparse MIL models in E2E are actually the primary bottleneck hindering the performance improvement**. We define this as the E2E optimization challenge, with **its** **mathematical definition provided in the Supplemental Material**. This differs entirely from the focus of previous E2E papers, which mainly concentrated on improving sampling strategies and reducing memory usage, yet still showed significant performance gaps compared to foundation model-based two-stage schemes.
>
> 【**Architectural novelty**】Based on this E2E optimization challenge, we propose a concise yet effective MIL model, ABMILX. Therefore, **the design motivation of this model differs from the common attention-based aggregators mentioned by the reviewer**, which focus on the two-stage paradigm with offline feature inputs. Additionally, **the design of the model is novel, and Reviewer igAe has expressed strong interest in the ABMILX structure**. Specifically, we introduce a multi-head mechanism to the classic sparse architecture ABMIL to alleviate the extreme sparsity of the original architecture. More importantly, **we utilize global feature correlation to refine local attention scores**. **This utilization of global correlation is pioneering—existing methods typically** **utilize global correlations to** **refine** **instance** **features rather than** **instance** **attention scores**. Furthermore, we suggest that compared to existing works, which often compute local attention and global attention independently, our idea of using global attention to refine the local one is also a quit ecreative attention combination manner.  For instance, as shown in our experimental results, RRT-MIL, which uses global attention and local attention independently, achieves significantly worse performance than our ABMILX in E2E learning. It is precisely the optimization challenge caused by sparse attention scores that motivated our innovation. **We provide rigorous and comprehensive proofs in the Supplemental Material regarding how these two mechanisms reduce optimization risk.**
>
> 【**Experimental and community contributions**】We have conducted a comprehensive and systematic evaluation of E2E and the proposed ABMILX, demonstrating the potential of E2E in computational pathology. Comparisons were performed across **7 datasets**, **4 different tasks**, **3 state-of-the-art foundation models** of varying sizes, and **9 different MIL aggregators**. **Without any pre-training on pathological data**, with a training cost of only **10 3090 GPUHours**, our method outperforms foundation models on multiple challenging benchmarks while maintaining a significant inference efficiency advantage. Furthermore, we supplemented results showing its **generalization comparable to foundation models**, as well as the **excellent performance of ABMILX on non-pathological datasets or other application settings**. More importantly, to demonstrate the potential of E2E, we supplemented **experiments on ABMILX and the proposed E2E framework in UNI PEFT**. Due to space constraints, details are provided in responses **R1 and R3 to Reviewer igAe**.
>
> > **W2&Q1: Ablation of ABMILX and MRIS**
>
> **R2:** In fact, we have already provided comprehensive and detailed ablation experiments in our paper (**Sec.3.3, 4.3, and Table 3 in main text**) regarding the performance contributions of ABMILX and MRIS. Here, we have reorganized and summarized a clearer version, as shown in the table below. Moreover, it should be clarified that we did not claim in the original paper that MRIS could address optimization risks. It is merely a sampling strategy that efficiently incorporates multi-scale information while keeping low time cost. And ABMILX is designed to mitigate the optimization risks in the E2E process.
>
> | Task/Method | ABMIL + RS | ABMIL + MRIS | ABMILX + RS | ABMIL+ MHLA+MRIS | ABMIL+ MHLA+ A+(ABMILX) + MRIS |
> | ----------- | ---------- | ------------ | ----------- | ---------------- | ------------------------------ |
> | Survival    | 62.64      | 62.70        | 67.24       | 63.80            | 67.78                          |
> | Subtyping   | 87.99      | 89.23        | 92.72       | 91.58            | 93.97                          |
> | Grading     | 73.51      | 75.46        | 76.77       | 77.56            | 78.34                          |
>
> > **W2: The comparison with sampling strategies**
>
> **R3:** In fact, our paper has already presented sufficient sampling strategies for comparison. These include naive **random sampling (RS)** and our MRIS, as well as **cluster sampling (C2C)**, **naive attention sampling (Attention)**, and **variational information bottleneck-based attention sampling (FT)** (which fall under selective sampling strategies). Here, we have reorganized and summarized a clearer version, as shown in the table below. We suggest these experiments validate one of our insights: in the E2E process, the role of MIL in optimization is often more pronounced than that of sampling strategies.
>
> | Task/Method | ABMILX + Attention | ABMILX + MRIS | ABMILX + RS | ABMIL + RS | ABMIL + MRIS | ABMIL + Attention | C2C   | FT    |
> | ----------- | ------------------ | ------------- | ----------- | ---------- | ------------ | ----------------- | ----- | ----- |
> | Survival    | 66.53              | 67.78         | 67.24       | 62.64      | 62.70        | 62.74             | -     | -     |
> | Subtyping   | 93.14              | 93.97         | 92.72       | 87.99      | 89.23        | -                 | 91.13 | 86.48 |
> | Grading     | 77.43              | 78.34         | 76.77       | 73.51      | 75.46        | -                 | 62.91 | 66.06 |
>
> > **Q2: Latest Baselines**
>
> **R4:** In fact, **we compared the performance of WIKG and RRT under FM features (UNI, CHIEF, GIGAP) in Table 1 of the Supplemental Material.** In the table below, we have reorganized these results and **supplemented 2DMamba [1]**. In the revised version, this content will be included in the main text.
>
> | Encoder   | Method  | Grad.     | Sub.      | Surv.     |
> | --------- | ------- | --------- | --------- | --------- |
> | R18 (E2E) | ABMILX  | 78.34     | 93.97     | **67.78** |
> | R50 (E2E) | ABMILX  | **78.83** | **95.17** | 67.20     |
> | UNI       | RRT-MIL | 74.00     | 94.61     | 66.91     |
> | UNI       | WIKG    | 74.97     | 94.19     | 65.28     |
> | UNI       | 2DMamba | 74.97     | 93.59     | OOM       |
> | CHIEF     | RRT-MIL | 69.73     | 92.49     | 67.30     |
> | CHIEF     | WIKG    | 71.22     | 93.13     | 65.92     |
> | CHIEF     | 2DMamba | 71.59     | 90.96     | OOM       |
> | GIGAP     | RRT-MIL | 72.46     | 94.84     | 65.40     |
> | GIGAP     | WIKG    | 72.17     | 94.76     | 66.97     |
> | GIGAP     | 2DMamba | 75.36     | 93.33     | OOM       |
>
>  *OOM = Out-of-Memory on 24GB-RTX-3090*
>
> > **Q3: optimization collapse**
>
> **R5:** First, as stated in Section 3.3 of the main text, **we have provided additional mathematical analysis on optimization collapse in the supplemental material (Sec.A)**, which will be incorporated into the main text in revised revisions. A concise mathematical explanation is that during the E2E process, the gradients received by the encoder $\theta$ for $i$-th instances consists of the gradients from the instance embedding $e_i$ and the gradients from the corresponding normalized attention score $\hat{a}_i$:
>
> $\theta \leftarrow \frac{\partial \mathcal{L}}{\partial e_i}+\frac{\partial \mathcal{L}}{\partial \hat{a}_i}=\frac{\partial \mathcal{Z}}{\partial e_i}\cdot\frac{\partial \mathcal{L}}{\partial Z}+\frac{\partial \mathcal{L}}{\partial \hat{a}_i}= \frac{\partial \mathcal{\sum\hat{a}_j e_j}}{\partial e_i}\cdot\frac{\partial \mathcal{L}}{\partial Z}+\frac{\partial \mathcal{L}}{\partial \hat{a}_i}=\hat{a}_i\frac{\partial \mathcal{L}}{\partial Z}+\frac{\partial \mathcal{L}}{\partial \hat{a}_i}$,
>
> where $ \mathcal{L}$ denotes the loss value and $Z=\sum\hat{a}_je_j$ denotes the aggregation of all instance embeddings. It could be concluded that the gradients from $i$-th instance embedding $\frac{\partial \mathcal{L}}{\partial e_i}=\hat{a}_i\frac{\partial \mathcal{L}}{\partial Z}$ are scaled by corresponding normalized attention score $\hat{a}_i$. And sparse attention tends to assign high scores to noisy instances $\mathcal{N}$, leading to optimization collapse. Therefore, we suggest using the maximum attention score of noisy instances as a metric to quantify optimization collapse $\mathcal{R}$:
>
> $\mathcal{R} = O(\max_{i \in \mathcal{N}} \hat{a}_i).$
>
> To better demonstrate optimization collapse, we use the product of the maximum attention score of noisy instances for each slide and the total number of instances as the final metric (**MAX-N**). Furthermore, we have measured MAX-N on the CAMELYON dataset in the early training stage. The results are reported in the table below, demonstrating that ABMILX effectively mitigates optimization collapse while maintaining reasonable sparsity. These experiments and analyses will also be added to the revised version. **Finally, in page 6 of the main text, we have demonstrated how ABMILX mitigates optimization collapse through visualization.**
>
> | Metric/Method | ABMIL   | ABMILX |
> | ------------- | ------- | ------ |
> | MAX-N         | 21.2162 | 2.6557 |
> | Sparsity      | 80      | 36     |
> | Performance   | 91.78   | 95.88  |
>
> *Sparsity is a metric proposed in the main text to measure model sparsity, and it can also be regarded as an indirect metric of optimization collapse.*
>
> [1] 2DMamba: Efficient State Space Model for Image Representation with Applications on Giga-Pixel Whole Slide Image Classification. CVPR 2025.

---

> > ### Author Response · Authors · 2025-08-05
> >
> > Dear Reviewer tziM,
> >
> > Thank you once again for your valuable comments on our submission. **As the discussion phase is approaching its end, we would like to kindly confirm whether we have sufficiently addressed all of your concerns (or at least part of them)**. Should there be any remaining questions or areas requiring further clarification, please do not hesitate to let us know. If you are satisfied with our responses, we would greatly appreciate your consideration in adjusting the evaluation scores accordingly.
> >
> > We sincerely look forward to your feedback.

---

> ### Comment · Area_Chair_ptkx · 2025-08-07
> **Reviewer tzIM - Please read over the other reviews and respond to the authors**
>
> See title. Please acknowledge that you have read through things and respond to the authors.

---

> ### Author Response · Authors · 2025-08-08
>
> Dear Reviewer tziM,
>
> **With the discussion phase ending soon**, we wish to ensure we have adequately addressed all your valuable concerns. Your initial feedback was instrumental in improving our work, and we are eager to know if our responses and the updated manuscript meet your expectations. Please let us know if you have any remaining questions. Should our response resolve your primary concerns, **we would be grateful if you would consider updating your evaluation**.
>
> Thank you for your valuable time.

---

> ### Author Response · Authors · 2025-08-08
> **Official Comment by Authors**
>
> Dear Reviewer tziM,
>
> As the discussion phase draws to a close, we are writing to make one final check-in. Your feedback has been critical in shaping our revisions, and in our rebuttal, we’ve worked diligently to address each point you raised. We sincerely look forward to your corresponding conscientious feedback of high-quality to help us polish our work while helping the Area Chair assess the work with full context. We understand your schedule is demanding and deeply value the time you’ve already invested. And If our revisions have met your expectations, we would appreciate your consideration in updating your evaluation.
>
> Thank you again for your valuable time.

---

### Official Review · Reviewer_AkkT · 2025-07-04

**Clarity:** 3
**Significance:** 3
**Originality:** 3
**Rating:** 4
**Confidence:** 2

**Summary:**

This paper revisits end-to-end (E2E) learning with slide-level supervision for computational pathology, aiming to overcome the limitations of the traditional two-stage paradigm. The authors identify a key optimization challenge caused by sparse-attention MIL modules and introduce ABMILX, a novel MIL method that combines multi-head attention and global correlation refinement. Their proposed E2E pipeline with ABMILX achieves state-of-the-art performance while being significantly more computationally efficient than existing foundation models. This work demonstrates the potential of E2E learning in pathology and encourages further research in this direction.

**Questions:**

1. How sensitive is ABMILX to choices like the number of attention heads and the global attention scaling factor, and how do these impact its performance?

2. Can the proposed E2E pipeline handle real-world variations such as slide size, noise, and rare classes not present in benchmark datasets?

3. Are there any noticeable gaps between the theoretical analysis and the experimental results, and if so, what further work is planned to address them?

**Ethical Concerns:**

["NO or VERY MINOR ethics concerns only"]

**Final Justification:**

Thank you for the detailed rebuttal and the added experiments/clarifications. I will keep my score. I have also considered the other reviewers’ comments and your responses during the discussion. In brief: strong empirical gains and practical efficiency; architectural novelty feels moderate.

**Limitations:**

Yes

**Quality:**

3

**Strengths And Weaknesses:**

### Strengths

1. Comprehensive Analysis of Optimization Challenges
The paper offers a detailed analysis of why E2E learning underperforms in CPath due to MIL-induced attention sparsity and clearly identifies the core bottleneck.

2. Novel and Effective MIL Architecture
The proposed ABMILX module successfully mitigates the limitations of sparse attention, leading to clear performance improvements in E2E learning.

3. Efficiency and Clinical Relevance
The E2E pipeline provides competitive or superior accuracy to state-of-the-art models at a fraction of the computational cost, making it more practical for real-world clinical use.


### Weaknesses
1. Limited Theoretical Justification
The theoretical foundations of ABMILX are only partially provided, with formal proofs or rigorous analysis not fully developed in the main text.

2. Generality Not Fully Demonstrated
Experiments are focused on a few public cancer datasets, so it is unclear how well the method generalizes to other pathology domains or more diverse clinical cases.

3. Limited Discussion on Generalization of Large FMs
The comparison with large foundation models is mostly centered on accuracy and efficiency, with little discussion of their generalization or transferability strengths.

---

> ### Author Rebuttal · Authors · 2025-07-31
>
> Thanks for providing feedback and taking the time to review our work! We have provided point-by-point clarifications on the issues and supplemented the experiments as suggested.
>
> > **W1: Clarification on Theoretical Justification**
>
> **R1:** As stated in Section 3.3 of the main text, we have provided a comprehensive and rigorous mathematical analysis of ABMILX in the Supplemental Material (Sec.A). This includes the definition of optimization risk, as well as complete proofs demonstrating how the proposed Multi-head Local Attention Mechanism and Attention Plus Propagation reduce optimization risk. It will be included in the main text in the revised version.
>
> > **W2&Q2: The results of more medical tasks and clarification on real-world data**
>
> **R2:**
>
> - **【Clarification on dataset used】** First, we would like to clarify that computational pathology is closely linked to cancer analysis, as the **analysis of pathological images is typically regarded as the gold standard for cancer diagnosis and prognosis in clinical practic**e. Therefore, cancer datasets are commonly used in the field of computational pathology. Due to ethical concerns and the inaccessibility of private data, we utilized public datasets. However, the public datasets we used are **diverse in sources** (TCGA, CAMLYON, PANDA), **covering various cancers across multiple organs** (prostate, breast, lung, bladder) with **rich and challenging task types** (cancer subtyping, cancer grading, cancer patient survival time analysis, cancer metastasis detection). These datasets are **derived from real-world clinical scenarios**. For instance, TCGA, the largest cancer database in the United States, is not a benchmark specifically designed for model evaluation. Furthermore, **these datasets vary in size** (PANDA images typically have dozen of millions of pixels, while TCGA images usually exceed 1 billion pixels) and **noise ratios** (CAMLYON has less than 10% cancer cell area, whereas other datasets generally have over 40%). In addition, we **supplemented external validation experiments using CPTAC data (another large cancer data source)** to demonstrate the performance of our method across more clinical cases. The response to **R3** provides specific results and details.
>
> - **【Diabetic retinopathy grading】** Furthermore, we further verify the generalizability of our method to other high-resolution E2E medical tasks through diabetic retinopathy grading (DR) [1] and existing frameworks [2]. The results are reported in the table below, which demonstrate the advantages of the proposed ABMILX in optimizing high-resolution end-to-end tasks.
>
>   - |          | ABMIL | TransMIL | ABMILX |
>     | -------- | ----- | -------- | ------ |
>     | DR (Acc) | 75.3  | 74.3     | 78.1   |
>
> > **W3: The generalization compared foundation models**
>
> **R3:** We have supplemented the following external validation on CPTAC to demonstrate the generalization ability of the encoder trained on TCGA via E2E learning:
>
> | TCGA->CPTAC          | R50+ABMIL | R50+TransMIL | R50+WIKG | UNI+ABMIL | UNI+TransMIL | UNI+WIKG | R50+ABMILX (E2E) |
> | -------------------- | --------- | ------------ | -------- | --------- | ------------ | -------- | ---------------- |
> | CPTAC-NSCLC (AUC)    | 66.42     | 74.59        | 64.04    | 83.73     | 85.24        | 83.56    | **85.19**        |
> | CPTAC-LUAD (C-index) | 46.34     | 48.24        | OOM      | 53.59     | 51.36        | OOM      | **54.00**        |
>
> *OOM = Out-of-Memory on 24GB-RTX-3090*
>
> The experimental results demonstrate that the R50 optimized via an E2E approach on TCGA not only exhibits significant improvements in generalization but even outperforms UNI (a ViT-L model pre-trained on 1B-scale pathological data). This validates the strengths of both E2E learning and our proposed ABMILX in terms of generalization ability and transferability across real-world variations.
>
> > **Q1: Clarification on hyperparameters**
>
> **R4:** We analyzed the impacts of different global attention scaling factors and numbers of attention heads in **Section 3.3 of the main text** and **Section C.4 of the Supplemental Material**, respectively. We have reorganized these experiments in the table below:
>
> | Global Attention Scaling Factors | Subtyping | Survival |
> | -------------------------------- | --------- | -------- |
> | $\alpha = 0$                     | 91.58     | 63.80    |
> | $\alpha = 1$                     | 92.84     | 65.49    |
> | Learnable $\alpha$               | 93.97     | 67.78    |
>
> | Numbers of Attention Heads | Grading | Subtyping | Survival |
> | -------------------------- | ------- | --------- | -------- |
> | 4                          | 78.34   | 91.65     | 62.83    |
> | 8                          | 76.48   | 93.97     | 67.78    |
> | 16                         | 76.50   | 92.59     | 64.62    |
>
> We use a learnable global attention scaling factor by default, while the number of heads has an impact on model performance. However, selecting 8 or 4 by default can achieve sufficiently strong performance without the need for additional hyperparameter tuning.
>
> > **Q3: Gaps between the Theoretical Analysis and the Experimental Results**
>
> **R5:** According to our theoretical analysis in the Supplemental Material, the maximum attention score of noisy instances can serve as a representative of optimization risks. And our theoretical analysis suggests that ABMILX can mitigate this risk while maintaining reasonable sparsity. **Therefore,** **we use the product of the maximum attention score of noisy instances for each slide and the total number of instances (MAX-N)** **to validate the gap between theoretical analysis and the experimental results****.** And we have measured MAX-N on the CAMELYON dataset in the early E2E training stage. The results are reported in the table below. As consistent with the theoretical analysis, ABMILX indeed mitigates optimization risks effectively while maintaining reasonable sparsity. **This proves that there is no significant gap between our theoretical analysis and experimental results. These experiments and analyses will also be added to our revised version.** A minor gap is that our experimental results have shown that the value of $\alpha$ has a certain impact on the final performance, but this was not fully discussed in our theoretical analysis. We suggest that the influence of the value of $\alpha$ stems from its role in controlling the attention sparsity of MIL. In the future, we will conduct corresponding theoretical analyses and experiments to design a MIL architecture with better performance in terms of sparsity.
>
> | Metric/Method | ABMIL   | ABMILX |
> | ------------- | ------- | ------ |
> | MAX-N         | 21.2162 | 2.6557 |
> | Sparsity      | 80      | 36     |
> | Performance   | 91.78   | 95.88  |
>
> [1] Diagnostic assessment of deep learning algorithms for diabetic retinopathy screening. Information Sciences.
>
> [2] No Pains, More Gains: Recycling Sub-Salient Patches for Efficient High-Resolution Image Recognition. CVPR 2025.

---

> > ### Comment · Reviewer_AkkT · 2025-08-05
> >
> > Dear authors,
> >
> > Thank you for your detailed rebuttal and comprehensive experimental analysis. I am especially impressed by the strong generalization results, showing that even when training separate models for each task, your E2E approach sometimes generalizes better than large foundation models.
> >
> > This brings up an interesting question: if your E2E framework were extended to multi-task learning—i.e., training on multiple tasks jointly—do you think it could serve as a true foundation model for computational pathology, potentially matching or even surpassing large-scale self-supervised pre-trained models? Moreover, would it be feasible in such a case to obtain strong downstream performance on new tasks via simple linear evaluation, without finetuning?

---

> > > ### Author Response · Authors · 2025-08-05
> > >
> > > We sincerely appreciate your valuable suggestions and appreciation. Your proposed comment has deeply inspired us, as it outlines a more concrete direction for the large-scale E2E training we plan to conduct in the future.
> > >
> > > **We believe that multi-task E2E supervised pre-training holds the potential to surpass self-supervised pre-trained foundation models.** This is because the joint supervision of diverse tasks helps the model comprehensively understand pathological images, fully exploit data potential, and endow it with genuine zero-shot capabilities (eliminating the need for additional training). In fact, a more reasonable approach is to perform multi-task E2E supervised fine-tuning building upon self-supervised pre-trained foundation models—aligning with the paradigm evolution in foundational fields, such as from DINO to SAM and from GPT to ChatGPT. Moreover, **we conducted a preliminary attempt at UNI by applying ABMILX for E2E Parameter-Efficient Fine-Tuning (PEFT) on UNI.** The following results demonstrate the advantages of ABMILX in E2E tasks and preliminarily verify the potential of E2E to further enhance FMs. We note that work [1] adopted a similar approach but overlooked the impact of optimization challenges caused by MIL on E2E training.
> > >
> > > |       | UNI   | R50 (E2E) | UNI+PEFT (E2E) | UNI+PEFT (E2E) | UNI+PEFT (E2E) |
> > > | ----- | ----- | --------- | -------------- | -------------- | -------------- |
> > > |       | ABMIL | ABMILX    | ABMIL          | TransMIL       | ABMILX         |
> > > | PANDA | 74.69 | 78.83     | 74.11          | 76.61          | 80.99          |
> > >
> > > Due to constraints in computational resources and discussion time, we cannot provide additional experimental validation here. However, we commit to conducting relevant research in the future to advance the development of computational pathology. We once again express our gratitude for your patience, appreciation, and suggestions, and sincerely hope you can raise your evaluation score to support our continued research on E2E computational pathology.
> > >
> > > [1] Unlocking adaptive digital pathology through dynamic feature learning.

---

> ### Author Response · Authors · 2025-08-08
>
> Dear Reviewer AkkT,
>
> **With the discussion period drawing to a close**, we would like to follow up and gently check if our rebuttal has addressed your concerns to your satisfaction. We are on standby to provide any further clarifications you might need. **If you are satisfied with our responses, we would be very grateful if you would consider updating your evaluation**.
>
> Thank you again for your time and valuable feedback. We look forward to hearing from you.

---

### Note · Authors · 2025-08-12

Dear reviewers and Area Chairs,

we suggest that we have fully addressed most of the issues raised by the reviewers during the rebuttal and discussion periods. Our final remarks are as follows:

- We clarified this work has provided **sufficient experiments and theoretical analysis** to demonstrate the optimization risks from MIL in E2E optimization, including the direct metric MAX-N supplemented during the rebuttal stage (Reviewers **AkkT** and **tziM**).
- Although generalization and scalability are not the core of this paper, we also provided verification experiments on the **UNI model (E2E PEFT) and external tests (from TCGA to CPTAC)** to preliminarily demonstrate the generalization and scalability of our method during the Rebuttal stage. As noted by Reviewer AkkT, the strong generalization results are quite impressed (Reviewers **AkkT** and **igAe**).
- Regarding the innovativeness and effectiveness of ABMILX, we proved through extensive experiments that ABMILX is more effective than other MIL methods in the E2E optimization of various computational pathology tasks and other clinical tasks, and provided corresponding theoretical analysis. In addition, we suggest that, as noted by the Reviewer igAe, the approach of using global attention to correct local attention in ABMILX is novel and pioneering in the field (Reviewer **tziM** and **AkkT**).
- In the supplementary materials and during the rebuttal stage, we supplemented the performance of advanced MIL methods under **FM features, such as WIKG, RRT, 2DMamba, and PathGNN**. Additionally, we summarized and verified the generality of the combination of ABMILX with various sampling methods, including RS, MRIS, Attention, and Dual-Buffer (Reviewers **tziM** and **igAe**).
- As stated in the title, this work aims to revisit the E2E in CPath, analyze the reasons for the limitations on its effectiveness: ignoring the impact of MIL on the optimization risks of E2E, provide simple but effective solutions: MRIS+ABMILX, and corresponding efficient and systematic code. Although there are still some issues to be explored and improved in the E2E field, we suggest that our conference paper meets the acceptance criteria.

**Reviewers AkkT and igAe** have responded positively to our clarifications and supplements. **Given the absence of Reviewer tziM during the discussion phase**, we sincerely hope that Area Chairs and reviewers revisit and carefully consider the contributions of this paper.

---

### Decision · Program_Chairs · 2025-09-17

**Decision:**

Accept (poster)

**Comment:**

The paper identifies that extreme sparsity in MIL attention causes optimization failures in end-to-end (E2E) pathology training. It introduces ABMILX, combining multi-head local attention and a global correlation refinement, and shows that with multi-scale random sampling, their method can match or surpass strong two-stage foundation model baselines while being far more computationally efficient.

Strengths: They effectively describe a plausible failure mode in E2E MIL and an effective fix that is easy to integrate.

Weaknesses: Comparisons should also clarify that FMs are typically SSL-pretrained on massive unlabeled slides (a key OOD-generalization advantage, which isn't really assessed here) and that your E2E uses ImageNet-pretrained ResNets (at least in experiments, although I don't see why it couldn't be used with models that are randomly initialized as is common practice with foundation models). The ImageNet initialization could be a potential confounding factor for some of the claims, especially regarding the amount of time required.

Discussion & Rebuttal: During the rebuttal, the authors strengthened the paper with additional baselines and added results. These additions largely resolved concerns about evaluation rigor.

Decision and rationale: Accept (poster). The paper provides a valuable insight into E2E MIL instability and demonstrates a simple yet impactful solution. The caveats mentioned in the weaknesses ought to be explicitly mentioned in the paper.

Note: The two-stage paradigm was NOT introduced by Lu et al. [30]; earlier work (e.g., Campanella et al., 2019 [3]) predates this claim. They used MIL to learn an encoder for tile embeddings, and then used an RNN as an aggregator for those tile embeddings. This ought to be corrected in the paper (there may be an even earlier reference).